# ARIA: AN AGENT FOR RETRIEVAL AND ITERATIVE AUTO-FORMALIZATION VIA DEPENDENCY GRAPH

**Hanyu Wang**[1*]   **Ruohan Xie**[1*]   **Yutong Wang**[1,2*]   **Guoxiong Gao**[1,2*]   **Xintao Yu**[2,3]
**Bin Dong**[4,5,6†]
[1]Peking University   [2]IQuest Research   [3]Renmin University of China
[4]Beijing International Center for Mathematical Research and the New Cornerstone Science Laboratory, Peking University
[5]Center for Machine Learning Research, Peking University
[6]Center for Intelligent Computing, Great Bay Institute for Advanced Study, Great Bay University
`wanghanyu2004@stu.pku.edu.cn, xieruohan@stu.pku.edu.cn`
`wangyutong25@stu.pku.edu.cn, samggx@stu.pku.edu.cn`
`RUCxintaoyu@outlook.com, dongbin@math.pku.edu.cn`

## ABSTRACT

Accurate auto-formalization of theorem statements is essential for advancing automated discovery and verification of research-level mathematics, yet remains a major bottleneck for LLMs due to hallucinations, semantic mismatches, and their inability to synthesize new definitions. To tackle these issues, we present Aria (**A**gent for **R**etrieval and **I**terative **A**utoformalization), a system for conjecture-level formalization in Lean that emulates human expert reasoning via a two-phase Graph-of-Thought process: recursively decomposing statements into a dependency graph and then constructing formalizations from grounded concepts. To ensure semantic correctness, we introduce **AriaScorer**, a checker that retrieves definitions from Mathlib for term-level grounding, enabling rigorous and reliable verification. We evaluate Aria on diverse benchmarks. On ProofNet, it achieves 91.6% compilation success rate and 68.5% final accuracy, surpassing previous methods. On FATE-X, a suite of challenging algebra problems from research literature, it outperforms the best baseline with 44.0% vs. 24.0% final accuracy. On a dataset of homological conjectures, Aria reaches 42.9% final accuracy while all other models score 0%.

## 1 INTRODUCTION

In recent years, Interactive Theorem Provers (ITPs) such as Coq (Barras et al., 1999), Isabelle (Paulson, 1994) and Lean (Moura & Ullrich, 2021) have become crucial ecosystems for formalized mathematics. Among these, Lean 4, together with its comprehensive library Mathlib (mathlib Community, 2020), is pioneering a new paradigm for formalization. However, the continuous growth of this ecosystem is always constrained by the immense manual effort and the deep expertise that formalization demands. To address this, the research community has turned to Large Language Models (LLMs) for auto-formalization the process of translating informal (or natural language) mathematical statements and proofs into their formal counterparts. While these two processes are interconnected, the accurate formalization of statements is the foundational first step. A correctly formalized statement is a prerequisite for any valid proof and, on its own, is a valuable asset to the mathematical ecosystem, enabling better search, integration, and verification. Thus, despite progress in proof automation (Ren et al., 2025; Chen et al., 2025), the fidelity of this initial statement translation remains a critical bottleneck. LLMs frequently generate formal statements that suffer not only from compilation errors but also from more insidious semantic flaws, a challenge that intensifies when formalizing more complex research or conjecture-level statements.

---

[*]Equal contribution.
[†]Corresponding author.

These foundational shortcomings manifest in several critical downstream failures. An unfaithful translation can derail large-scale data generation pipelines, wasting significant computational budgets on attempts to prove an incorrect premise. For instance, modern provers often decompose complex proofs into smaller, informal lemmas, which are then individually translated and proven. In this workflow, a single flawed translation of a lemma not only invalidates the entire proof structure but can also contaminate the datasets generated during this process, which are crucial for fine-tuning future models. Furthermore, as the research community pushes towards formal models that can autonomously explore conjecture-level problems, the inability to create and utilize the necessary, often unseen, premises (i.e. definitions, lemmas, theorems, etc.) becomes a critical roadblock. Any system lacking this capability is bound to fail at the outset of such ambitious tasks. In this work, we address these challenges by introducing a robust methodology to generate, iterate, and verify formal statements, tackling these foundational bottlenecks through automated structural reasoning.

One primary challenge stems from the static nature and inherent fallibility of an LLM's pre-trained knowledge. While foundational work has demonstrated the potential of LLMs up to undergraduate mathematics (Gao et al., 2024b; Wang et al., 2025), these methods exhibit critical failure modes when confronted with research-level statements, where LLMs are prone to hallucination and outdated pre-trained knowledge. They generate invalid codes with functions either non-existent in Mathlib, or incompatible with rapidly evolving library toolchains. To address this, we integrate a Retrieval-Augmented Generation (RAG) framework, grounding the formalization process by dynamically querying the most current version of the Mathlib library, mitigating the model's dependence on static knowledge and ensuring compatibility with the evolving toolchain.

Beyond the issue of knowledge retrieval, a more profound challenge lies in synthesis. Research-level mathematics fundamentally involves creating new mathematical objects and definitions, one-pass generation methods, even when augmented with retrieval, fail at this task because they cannot spontaneously synthesize definitions for concepts absent from existing libraries. To address this, we develop an agentic pipeline driven by a Graph-of-Thought (GoT) formalizing process. This approach emulates an expert mathematician's workflow by recursively decomposing dependencies of definitions until they are well-grounded, then synthesizes their formal statements in a bottom-up order until the primary target is formalized. To ensure the robustness of this process, a compiler-in-the-loop reflection mechanism is employed at each node.

Once a statement is generated and pass the compiler check, the ultimate challenge is to ensure its semantic correctness. While existing methods like LeanScorer (Xuejun et al., 2025) have advanced semantic checking by performing fine-grained comparisons, they fail to detect subtle definitive discrepancies between formal and informal terms due to reliance on superficial textual similarity. To overcome this limitation, we introduce AriaScorer, an enhanced semantic checker that incorporates a term-level grounding step. AriaScorer retrieves the authoritative definitions of all Lean terms from Mathlib and injects this formal context into the comparison process, enabling a more rigorous and accurate evaluation.

Equipped with this validated checker, we evaluated Aria's end-to-end performance on a suite of research-level datasets. We measure final accuracy, which we define as the proportion of the generated formalized statements that pass both compiler and semantic correctness checks. The results demonstrate a significant leap over prior work, with Aria achieving 68.5% on the ProofNet benchmark while also surpassing previous state-of-the-art models on others, including FATE-H (71.0% vs. 43.0%) and FATE-X (44.0% vs. 24.0%). Most notably, on a challenging set of real-world mathematical conjectures where all baseline models score 0%, Aria achieves a 42.9% success rate, demonstrating a unique capability for research-level formalization.

The main contributions of this paper are as follows:

- We introduce Aria, a statement auto-formalizer agent that emulates the human formalization process by integrating retrieval-augmented generation, graph-of-thought planning, and a compiler-guided self-reflection mechanism that is especially effective on conjecture-level problems.

- We develop a term-level grounded semantic scorer, AriaScorer, to detect subtle discrepancies between informal statements and Lean terms, and to accurately verify the mathematical correctness of formalizations.

- We achieve state-of-the-art performance with substantial improvements over previous methods, reaching 68.5% on ProofNet, 71.0% on FATE-H, 44.0% on FATE-X, and 42.9% on real-world conjectures proposed by mathematicians.

The remainder of this paper is structured as follows. Section 2 reviews related work. Section 3 details our proposed methodology, including Aria's architecture and its core components. Section 4 presents our experimental results and their analysis. Finally, Section 5 concludes the paper.

## 2 RELATED WORK

**Auto-formalization**    The rapid advancement of Large Language Models (LLMs) has catalyzed significant progress in auto-formalization. Early efforts demonstrated success by leveraging few-shot in-context learning (ICL) (Wu et al., 2022; Patel et al., 2024; Zhou et al., 2024). As the Lean community grew and its Mathlib library became more comprehensive, the availability of large-scale datasets enabled the development of specialized models through supervised fine-tuning (SFT) (Azerbayev et al., 2023; Jiang et al., 2023; Gao et al., 2024b; Wang et al., 2025). More recently, Reinforcement Learning (RL) has shown potential in mathematics and inference, and several works have leveraged RL training to enhance the quality of auto-formalization (Xuejun et al., 2025; Huang et al., 2025). In parallel, other methods have focused on enhancing the quality and reliability of the generation process itself. With the increasingly powerful search capabilities within the Lean ecosystem, Retrieval-Augmented Generation (RAG) has proven effective at providing models with relevant definitions and theorems from the extensive Mathlib library (Lu et al., 2025). Concurrently, novel methodologies like Process-Supervised Verification (PSV) leverage direct feedback from compilers to guide the model's learning process, significantly improving the correctness and reliability of the generated formalizations (Lu et al., 2024). Similarly, in the adjacent field of automated theorem proving, recent works (Thakur et al., 2024; Zhou et al., 2025; Chen et al., 2025) have demonstrated the efficacy of reflection mechanisms, enabling systems to iteratively critique and refine their reasoning strategies.

**Semantic Check**    As methods for statement auto-formalization have become more sophisticated and diverse, it is crucial to establish a credible way to evaluate the extent to which the formal statement preserves the mathematical meaning of its informal counterpart. Human experts can certainly provide reliable evaluations of consistency (Azerbayev et al., 2023), but as statements grow more complex, such evaluations become increasingly demanding. Consequently, perplexity (Wang et al., 2018) and BLEU (Wang et al., 2018; Azerbayev et al., 2023) have been used as proxy metrics. It is also common to use an LLM to back-translate valid formal statements into informal statements, and then employ another LLM to assess semantic preservation (Ying et al., 2024; Gao et al., 2024b; Liu et al., 2025b). Additionally, a combined structure of unanimous voting among LLM judges and validation by Lean experts has been introduced, serving as a reward signal during training (Wang et al., 2025). Moreover, subtask decomposition of informal statements has been considered, resulting in a more fine-grained filtering of incorrect formalizations (Xuejun et al., 2025). Recently, an automated neuro-symbolic method for determining the mathematical equivalence of two formal statements has been widely adopted. This approach establishes equivalence if and only if a formal proof can connect the two statements, by using semantic-preserving tactics (Liu et al., 2025a; Wu et al., 2025).

## 3 METHODOLOGY

This section details our methodology, which is comprised of two primary components. The overall pipeline is shown in Figure 1. Section 3.1 describes Aria's architecture, a structured pipeline designed to navigate the deep conceptual dependencies in conjecture-level mathematical statements. Then Section 3.2 presents our integrated semantic checker, which verifies whether the agent's output is faithful to the original mathematical intent.

### 3.1 THE GRAPH-OF-THOUGHT (GOT) AUTO-FORMALIZER PIPELINE

In this section, we detail the architecture of our agent, Aria. This architecture moves beyond the conventional approach of direct, single-step generation. These methods often fail when applied to

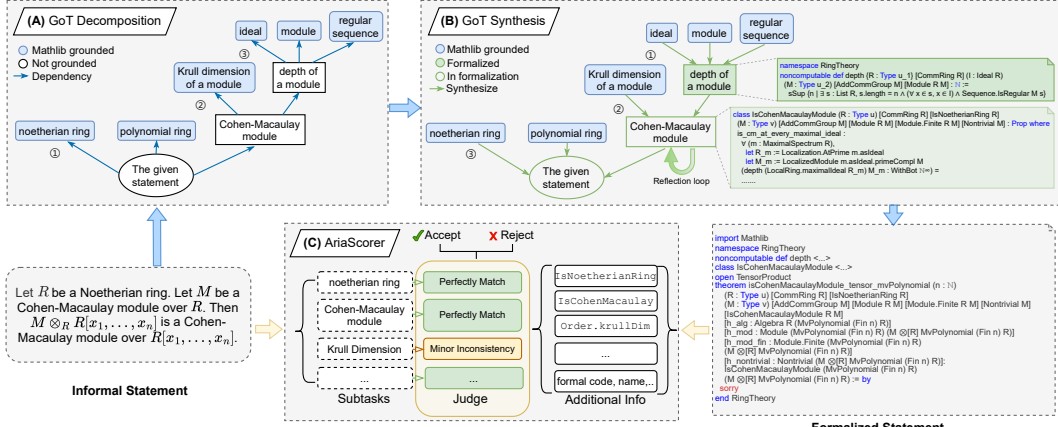

Figure 1: The overall pipeline of Aria system. (A) **Graph-of-Thought Decomposition:** Aria expands the informal statement into a dependency graph of concepts and grounds them in Mathlib. (B) **Graph-of-Thought Synthesis:** The system executes a bottom-up synthesis procedure guided by the graph, incorporating a self-reflection loop. (C) **AriaScorer:** A dedicated module that verifies the semantic correctness between the generated formal statement and the original informal statement.

complex, conjecture-level statements. As illustrated in Figure 1, our agent operates through a structured pipeline that systematically deconstructs, resolves, synthesizes and verifies a formalization, mirroring the methodical process of a human mathematician.

This pipeline uses a Graph-of-Thought (GoT) planner to deconstruct an informal statement into a conceptual graph, where each concept node represents a definition, structure or class, as illustrated in Figure 1 (A). Each concept node in the graph is then processed by a grounding module, which employs a Retrieval-Augmented Generation (RAG) framework powered by LeanSearch (Gao et al., 2024a) to anchor known concepts to the Mathlib library. For ungrounded concepts, a synthesis module generates new definitions bottom-up, as depicted in Figure 1 (B). All outputs are validated and refined by a compiler-in-the-loop reflection module. Finally, we employ a retrieval-based checker to verify semantic correctness.

### 3.1.1 GoT Decomposition Phase

To manage the complex, acyclic dependency graph of definitions and lemmas required to formalize a high difficulty-level statement, our agent's architecture is centered around a Planning Module based on the GoT paradigm. This module transforms the monolithic task of formalization into a structured, manageable workflow by modeling it as the construction and resolution of a conceptual dependency graph, as shown in Figure 1 (A). This approach is founded on a key principle of mathematical abstraction, which our agent leverages directly: any concept, no matter how complex, can be defined solely in terms of its immediate prerequisite concepts.

The core of our planning module is a conceptual dependency graph, a dynamic data structure that serves as the agent's working memory. This graph consists of concept nodes and directed edges, where each node represents a mathematical concept required for the final formalization.

For a given statement, Aria initiates a full formalization routine: it performs a top-down dependency expansion of the concept graph until all leaf nodes can be grounded in Mathlib. To achieve this grounding, the agent queries LeanSearch at each node. LeanSearch is a specialized search engine whose index is continuously updated to reflect recent versions of Mathlib, thereby remaining effective as Mathlib evolves. This retrieval process returns a ranked list of candidates from Mathlib, where each candidate consists of a formal statement and its corresponding informal description, ordered by their semantic relevance to the input concept name.

Since the top ranked search result is not always the canonical definition required for formalization, the agent employs an LLM as a sophisticated reasoner to analyze the retrieved candidates, identify-

ing the single best appropriate canonical definition for the concept. If the reasoner concludes that no suitable match exists among the candidates (i.e., the concept is not grounded in Mathlib), the node is treated as an internal node in the dependency graph (as depicted in Figure 1 (A). Its unresolved status triggers the planner to continue the top-down expansion of its children, after which the node is marked for synthesis.

### 3.1.2 GoT SYNTHESIZING PHASE

Immediately upon completing all expansions, the agent transitions to a bottom-up synthesis phase for the whole graph, which is shown in Figure 1 (B). The synthesis module is invoked for any concept that could not be grounded in the Mathlib library (for instance, the concept "Cohen-Macaulay Module" in Figure 1 (B)). This module is responsible for generating verifiable formal definitions from the ground up, guided by a robust compiler-in-the-loop reflection process that ensures syntactic correctness.

For a given target concept, the agent first collects the verified formal code of all its immediate dependencies (i.e. its children in the dependency graph) to use as context for the LLM to generate a formal Lean definition for the target. The generated code is immediately sent to the Lean compiler for a syntactic check. If compilation fails, the error message along with the failed code is then returned to the LLM as feedback for refinement. If it succeeds, the code is marked as synthesized and used for the synthesis of its parent node.

While this process ensures syntactic validity, it cannot preclude "correctly-typed but semantically wrong" translations. To check the semantic correctness of our code with a more flexible approach, our methodology incorporates an enhanced retrieval-based semantic consistency checker, which is detailed in Section 3.2.

### 3.2 SEMANTIC CORRECTNESS MODULE: ARIASCORER

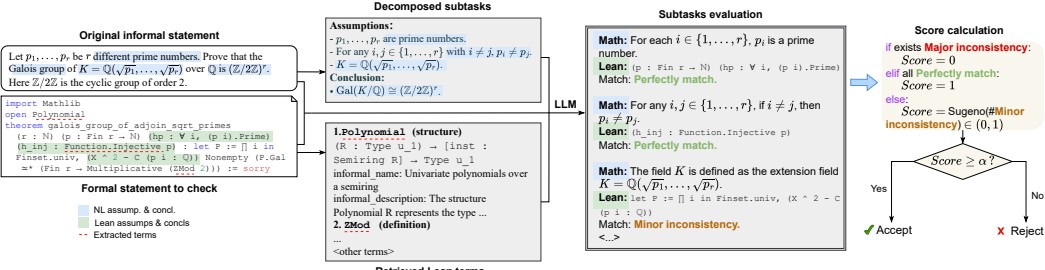

Figure 2: The overall pipeline of AriaScorer: informal statements are decomposed into subtasks, grounded with retrieved Lean terms, and their evaluations are aggregated into a final score, which is compared against a threshold $\alpha \in [0, 1]$ to yield a binary decision.

### 3.2.1 GROUNDWORK: LEANSCORER

We propose a semantic correctness checker for auto-formalized Lean statements aimed at mitigating hallucinations and reducing the false positives inherent in LLM-generated outputs. To address the densely packed, assumption-sensitive nature of high difficulty-level statements (such as conjectures), we adopt the subtask decomposition strategy of LeanScorer (Xuejun et al., 2025), which evaluates the semantic correctness through subtask-level comparisons.

Given an original informal statement, it is decomposed into atomic assumptions and conclusions by an LLM, and each resulting subtask is then evaluated to determine how well its formal clause matches the corresponding informal one. Subtasks are labeled as Perfectly Match, Minor Inconsistency, or Major Inconsistency, and these labels are aggregated via a fuzzy integral into a final score between $0$ and $1$, where $0$ indicates the presence of a major error and $1$ reflects perfect alignment across all subtasks. Besides these two cases, the score decays gradually from $1$ with accumulating minor inconsistencies, capturing the cumulative effect of subtle deviations. A tunable threshold is

applied to make binary decisions, balancing tolerance for small deviations against the need to reject semantically incorrect formalizations. Nonetheless, because this method still relies heavily on surface-level textual similarity, it remains vulnerable to semantic mismatches hidden beneath superficially close expressions, which motivates our introduction of a term-grounded evaluation module.

### 3.2.2 Term-level Semantic Grounding

To ensure alignment between the evaluation process and the true semantics of formal Lean statements, we introduce a new step: **term-level retrieval and interpretation**. In this step, we use jixia[1], a static analyzer for Lean, which extracts every Lean term referenced in the formal statement and queries the curated informalized Mathlib dataset established in Herald (Gao et al., 2024b) to retrieve each term's name, kind, type, value, informal name, and informal description. The retrieved term information, together with the original informal and formal statements, the decomposed subtask list, and few-shot examples, is then provided as context to the LLM during the subtask evaluation stage.

This process serves as the foundation for **semantic grounding**, enabling AriaScorer to reason over the true meanings of formal components rather than their surface syntax. As a result, AriaScorer can identify subtle inconsistencies, such as reversed parameter order or unintended type coercions, all of which are easily missed by purely textual comparison. This step helps prevent common LLM failure modes, including: (i) overlooking implicit preconditions or constraints embedded in Lean term definitions; (ii) misinterpreting Lean definitions by defaulting to their more familiar mathematical counterparts when the two diverge; and (iii) hallucinating incorrect explanations of Lean terms. These error types and how AriaScorer addresses them are discussed in Section 4.3.3, with detailed illustrations provided in the case studies.

By grounding evaluation in the actual semantics of Lean terms, AriaScorer provides more reliable and fine-grained assessments, particularly in cases involving newly introduced or user-defined structures. To validate the impact of this semantic grounding step, we present an ablation study in Section 4.3, showing clear gains in error detection and reductions in false positives.

## 4 Experiments

We conduct extensive experiments to evaluate the performance of Aria and AriaScorer. In Section 4.1, we describe the experimental setup of Aria, while Section 4.2 presents the main results. Section 4.3 demonstrates the comprehensive experiment to validate AriaScorer. Finally, we analyze the contributions of key components through ablation studies in Section 4.4.

### 4.1 Experimental Setup of Aria

This section outlines the experimental framework for rigorously evaluating Aria's performance, including the datasets used and the baselines for comparison.

### 4.1.1 Benchmarks

To rigorously assess our agent across diverse difficulty levels and problem types, we evaluate it on a suite of benchmarks. Specifically, we use the widely adopted ProofNet (Azerbayev et al., 2023) to ensure generalizability and comparability with existing work, and the FATE (Jiang et al., 2025) (Formal Algebra Theorem Evaluation) collection together with a dataset of 14 real conjectures to test performance on complex, research-level problems.

**ProofNet** To assess generalizability, we use ProofNet, a widely-adopted benchmark of undergraduate-level mathematics. This ensures our agent's sophisticated architecture is not only effective for complex conjectures but also robust and competitive on standard problems.

**FATE-H & FATE-X** The FATE collection tests our agent on advanced mathematics. FATE-H comprises problems from algebra final exams, while FATE-X contains more difficult problems from PhD qualifying exams and research literature. These benchmarks were specifically chosen to evaluate our agent's capabilities on complex, research-level mathematics.

---

[1] https://github.com/frenzymath/jixia

Table 1: End-to-end auto-formalization results comparing Aria against specialized models. All values are success rates (%); we report Compiler success rate and the stricter Final accuracy (passing both compilation and our AriaScorer semantic check). Results for the Conjectures dataset were manually verified. Kimina's score on ProofNet is marked due to potential data contamination[*].

| Method | ProofNet | | FATE-H | | FATE-X | | Conjectures |
|---|---|---|---|---|---|---|---|
| | Compiler | Final acc. | Compiler | Final acc. | Compiler | Final acc. | |
| **Aria** | **91.6** | **68.5** | 89.0 | **71.0** | 69.0 | **44.0** | **42.9** |
| Goedel-V2 (pass@16) | – | – | 77.0 | – | 37.0 | – | 0 |
| Goedel-V2 (pass@32) | – | – | 82.0 | – | 49.0 | – | 0 |
| Goedel-V2 (pass@64) | – | – | 88.0 | – | 58.0 | – | 0 |
| Goedel-V2 (pass@128) | – | – | **91.0** | 43.0 | 63.0 | 24.0 | 0 |
| Gemini-2.5-Pro (pass@1) | 55.8 | 27.8 | 35.0 | 31.0 | 27.0 | 21.0 | 0 |
| Goedel-V2 (pass@1) | 59.6 | 32.0 | 35.0 | 27.0 | 27.0 | 16.0 | 0 |
| Kimina (pass@1) | 70.4[*] | 24.7[*] | 10.0 | 0.0 | 5.0 | 1.0 | 0 |
| Herald (pass@1) | 48.5 | 18.3 | 24.0 | 12.0 | 8.0 | 5.0 | 0 |

[*] Kimina was trained on the ProofNet dataset, so its reported score may not reflect true generalization capabilities.

**Homological Conjectures in Commutative Algebra (Conjectures)** Finally, we test Aria on a set of 14 real-world Homological Conjectures (Wikipedia contributors, 2025) in Commutative Algebra, compiled by Melvin Hochster. These conjectures probe deep connections between the homological properties of a commutative Noetherian ring and its structural characteristics. This serves as a direct and challenging testbed of Aria's ability to formalize active mathematical research problems.

### 4.1.2 BASELINE MODELS

To evaluate the efficacy of our agent's architecture, we compare it against several leading statement auto-formalization models, including a powerful reasoning model Gemini-2.5-Pro (Google DeepMind) and specialized auto-formalizers including Goedel-Formalizer-V2-32B (Goedel-V2) (Lin et al., 2025), Kimina-Autoformalizer-7B (Kimina) (Wang et al., 2025) and Herald-translator (Herald) (Gao et al., 2024b).

### 4.2 MAIN RESULTS AND ANALYSIS

To evaluate the performance of our model, we conducted comprehensive tests comparing Aria with the baselines on benchmarks detailed in Section 4.1. As shown in Table 1, our agent demonstrates outstanding performance across all evaluations.

As shown in Table 1, Aria demonstrates a significant advantage over all baselines on each benchmark. However, Our GoT and reflection mechanisms require multiple LLM calls for each translation task within Aria. To ensure a fair comparison of efficiency, it is crucial to consider not only the success rate but also the computational cost, for which we use the number of API calls per problem as the primary metric. As Goedel-V2 is the top-performing specialized model at a single pass, with results comparable to the Gemini-2.5-Pro baseline, we select it for a direct comparison of computational budget against Aria. We first determined that Aria requires an average of 17.7 calls per problem on the FATE-X benchmark.

Based on this, we designed a series of experiments for Goedel-V2, ranging from pass@16 to pass@128. As shown in Table 1, while Goedel-V2's compilation rate scales with the number of calls, its final accuracy remains lower than Aria's. Aria maintains a higher final accuracy even when compared to Goedel-V2 at pass@128 (using more than 7x calls).

Most importantly, our comparative analysis on the Conjectures dataset reveals why Aria achieves its breakthrough performance. Through comprehensive case study of the generated codes, We identify distinct shortcomings in baseline models: large reasoning models tend to hallucinate incorrect interfaces due to insufficient expert knowledge of Mathlib, while specialized auto-formalizers lack the mathematical reasoning power to manage conjecture-level conceptual dependencies, as evidenced

by their tendency to simply replicate training data formats without a true understanding of the underlying mathematical logic. Aria's architecture, integrating GoT and retrieval module on top of a strong reasoning model, successfully addresses both limitations. We provide case studies of formalized conjectures in Appendix A for further illustration.

## 4.3 Validation of AriaScorer

### 4.3.1 Experimental Setup

We validated our semantic correctness checker against leading alternatives on the FATE-X benchmark. The evaluation used the Aria agent's syntactically correct, auto-formalized outputs. This benchmark contains complex mathematical statements and advanced definitions, providing a rigorous test of semantic precision.

**Ground truth dataset construction** We create an expert-validated ground truth dataset by labeling each formalization as "True" or "False" based on its mathematical fidelity. The annotations are provided by an algebra Ph.D. candidate in pure mathematics and has also contributed to Mathlib, then independently verified by a second expert with the same credentials. We then used this dataset to benchmark the performance of several semantic correctness checkers.

**Baselines** We benchmark AriaScorer against several established methods for checking semantic correctness. The first is LeanScorer (Xuejun et al., 2025), a method using decomposition and matching, which we re-implemented as its original version is not open-source. Our re-implementation of LeanScorer also serves as a critical ablation study for AriaScorer, representing our full pipeline but without the term-level grounding step. The second is Back Translation (Ying et al., 2024; Gao et al., 2024b), a widely-used pipeline that translates a formal statement back to an informal one and uses an LLM to judge the similarity. For a controlled comparison, AriaScorer, LeanScorer, and BackTranslation are all built upon the same base model: Gemini-2.5-Pro. We also evaluate Gemini-2.5-Pro's performance on this task directly. This comparison framework ensures that AriaScorer's accuracy improvements can be attributed specifically to our novel term-level analysis, rather than the underlying language model.

**Evaluation Metrics** We evaluate performance using binary classification, where formalizations are labeled positive (correct) or negative (flawed). Performance is based on the counts of True Positives (TP), True Negatives (TN), False Negatives (FN), and False Positives (FP). A False Positive, for instance, occurs when a checker incorrectly approves a flawed formalization. These four outcomes are then used to calculate and report accuracy, precision, recall, and F1 score.

### 4.3.2 Performance of AriaScorer

Table 2: Performance comparison of distinct semantic correctness checkers. It is carried out on the auto-formalized output of Aria on FATE-X. The score threshold for binary decision is denoted as $\alpha$.

| | AriaScorer ($\alpha = 0$) | AriaScorer ($\alpha = 0.9$) | LeanScorer ($\alpha = 0$) | LeanScorer ($\alpha = 0.9$) | Back Translation | Gemini |
|---|---|---|---|---|---|---|
| TP | 50 | 42 | 46 | 44 | 7 | 45 |
| TN | 12 | 15 | 3 | 7 | 16 | 8 |
| FP | 5 | 2 | 14 | 10 | 1 | 9 |
| FN | 2 | 10 | 6 | 8 | 45 | 7 |
| Accuracy | **89.9%** | 82.6% | 71.0% | 73.9% | 33.3% | 76.8% |
| Precision | 90.9% | **95.5%** | 77.6% | 81.5% | 87.5% | 83.3% |
| Recall | **96.2%** | 80.8% | 88.5% | 84.6% | 13.5% | 86.5% |
| F1 | **93.5%** | 87.5% | 82.1% | 83.0% | 23.3% | 84.9% |

AriaScorer is the top-performing model for semantic correctness checking on Aria's output from FATE-X. At a threshold of $\alpha = 0$, it achieves the highest accuracy (89.9%), recall (96.2%), and F1 score (93.5%), significantly outperforming all baselines. Its superior precision and recall compared to LeanScorer underscore the benefits of term-level grounding. Increasing the threshold to $\alpha =$

0.9 boosts AriaScorer's precision to a peak of 95.5%. This demonstrates a key trade-off: a lower threshold is more tolerant of mathematically equivalent forms, maximizing recall, while a higher threshold imposes stricter criteria, minimizing false positives for real-world deployment. In contrast, the Back Translation baseline, which demands an exact textual match, achieves very high precision but suffers from low overall recall. While we adopt the high-precision setting of $\alpha = 0.9$ in all other experiments, the results at $\alpha = 0$ best demonstrate the fundamental advantage of our term-grounded approach.

### 4.3.3 KEY FINDINGS OF ARIASCORER

By incorporating term-level grounding, AriaScorer addresses common failure modes in semantic correctness checking. Our ablation study highlights three of its key strengths:

**Implicit Semantic Inclusion**  By retrieving a formal term's full definition from Mathlib, AriaScorer identifies any implicit preconditions or constraints it contains. This uncovers crucial dependencies for accurate evaluation that purely textual comparisons would overlook. (See Appendix B.1).

**Definition Discrepancy Detection**  AriaScorer detects subtle discrepancies between a formal term's precise definition and the informal concept's intended meaning. By comparing the retrieved Mathlib definition against the original problem's context, it identifies when a Lean term, though textually similar, carries a different mathematical interpretation. (See Appendix B.2).

**Hallucination Suppression via Grounding**  AriaScorer suppresses LLM hallucinations by grounding the evaluation process. Before invoking the LLM, it injects the authoritative Mathlib definitions of all formal terms into the prompt. This constrains the model to reason based on verified ground truths, ensuring its output reflects the actual semantics of the formal code. (See Appendix B.3).

### 4.4 SUMMARY OF ABLATION STUDIES

We conduct a series of comprehensive ablation studies to quantify the unique contributions of Aria's core components: the Reflection mechanism, the Graph-of-Thought (GoT) planner, and the Retrieval-Augmented Generation (RAG) module. Our findings, particularly on the challenging Conjectures dataset, demonstrate that all three are indispensable.

- Ablating the Reflection module, caused performance to collapse on both FATE-X and Conjectures, proving its necessity for achieving correct codes.
- Removing the GoT planner crippled the agent's ability to formalize novel concepts, reducing successful conjectures from 6 to 1. This highlights its critical role in imposing logical structure. Moreover, we found that the impact of ablating the GoT module is more pronounced on more challenging datasets.
- Disabling the RAG module results in a complete 0% success rate on Conjectures, confirming its crucial function in grounding the agent and preventing foundational hallucinations of non-existent concepts.

Detailed procedures and analysis are provided in Appendix C.

## 5 CONCLUSION

In this paper, we present Aria, a statement auto-formalization agent integrating retrieval-augmented generation, graph-of-thought planning, and self-reflection mechanism. This integrated approach makes Aria the first agent capable of autonomously synthesizing the complex novel definitions required to formalize high difficulty-level mathematical statements such as conjectures. This capability directly addresses a core limitation of prior methods, which fail due to hallucination and logical errors when encountering unseen concepts. Moreover, we presented a novel semantic correctness checker, AriaScorer, that retrieves definitions from Mathlib for term-level grounding, enabling rigorous and reliable verification.

Our comprehensive experimental results demonstrate that our agent achieves leading final accuracy on benchmarks of varying difficulty, from the undergraduate level to conjectures. This success is particularly pronounced on the highly challenging Homological Conjectures dataset, where our agent achieves breakthrough performance.

Given that statement formalization is a critical prerequisite for theorem proving, our successful formalization of conjecture-level statements established a solid foundation for future work on automated mathematical proof at this frontier of research.

## ACKNOWLEDGEMENTS

This work is supported in part by National Key R&D Program of China grant 2024YFA1014000, Fundamental and Interdisciplinary Disciplines Breakthrough Plan of the Ministry of Education of China (JYB2025XDXM113), the New Cornerstone Investigator Program, and Ubiquant.

Xintao Yu is supported by the "Qiushi Academic-Dongliang" Project of Renmin University of China (No. RUC24OSDL015).

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

## A CASE STUDY FOR ARIA'S GENERATED STATEMENTS

In this section, we present a qualitative analysis of Aria's generated statements through several representative case studies to illustrate its strengths and limitations. For each case, we visualize the agent's conceptual dependency graph as a "blueprint"–a visualization style standard in the Lean community for representing dependencies–to illustrate its planning process. We then present the final formalization and compare it against the outputs generated by the Goedel-V2-Formalizer-32B and Gemini-2.5-Pro model for the same problem.

### A.1 EXAMPLE 1: KOETHE'S CONJECTURE

INFORMAL STATEMENT

Let $R$ be a ring. If $R$ has no non-zero nil ideal (two-sided), then it has no non-zero nil one-sided ideal (neither left nor right).

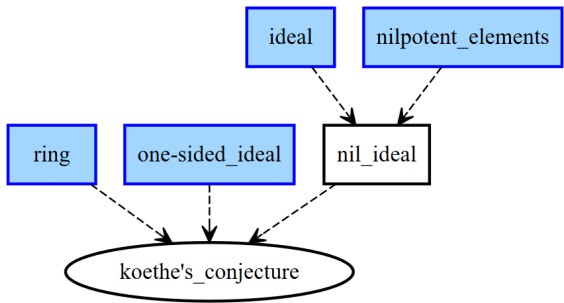

Figure 3: Dependency Graph of Koethe's Conjecture

ARIA GENERATED STATEMENT

```
import Mathlib

universe u

/-- An ideal `I` is a nil ideal if every element of `I` is nilpotent. -/
def IsNil {R : Type u} [Semiring R] (I : Ideal R) : Prop :=
  ∀ x ∈ I, IsNilpotent x

/--
If a ring `R` has no non-zero nil two-sided ideals, then it has no
    non-zero nil
one-sided ideals (neither left nor right).
-/
theorem no_nil_ideals_implies_no_nil_one_sided_ideals {R : Type u}
    [Ring R]
    (h : ∀ (I : Ideal R), IsNil I → I = ⊥) :
    (∀ (L : Submodule R R), (∀ x ∈ L, IsNilpotent x) → L = ⊥) ∧
    ∀ (J : Submodule (MulOpposite R) R), (∀ x ∈ J, IsNilpotent x) → J
    = ⊥ :=
  sorry
```

GOEDEL GENERATED STATEMENT

```
import Mathlib
import Aesop

set_option maxHeartbeats 0
```

```
open BigOperators Real Nat Topology Rat

theorem statement (R : Type*) [Ring R] :
  (∀ I : Ideal R, I ≠ ⊥ → ∃ x ∈ I, ¬IsNilpotent x) →
    (∀ L : Ideal R, L ≠ ⊥ → ∃ x ∈ L, ¬IsNilpotent x) ∧
      (∀ R : Ideal R, R ≠ ⊥ → ∃ x ∈ R, ¬IsNilpotent x) := by sorry
```

GEMINI GENERATED STATEMENT

```
import Mathlib

variable (R : Type*) [Ring R]

def koethe_conjecture : Prop :=
  (∀ (I : Ideal R), IsNil I → I = ⊥) →
  (
    (∀ (J : Submodule R R), IsNil J → J = ⊥) ∧
    (∀ (K : Submodule Rˢᵒᵖ R), IsNil K → K = ⊥)
  )
```

ANALYSIS

This example clearly reflects the architectural advantages of Aria over both the specialized formalization model like Goedel and a powerful general reasoning model like Gemini.

Among the three outputs, Gemini's results successfully preserves the original mathematical intent, correctly identifying the need to distinguish between two-sided and one-sided ideals. However, it fails to compile because it hallucinates a non-existent API (`IsNil`), reflecting a disconnect between its strong high-level reasoning and its knowledge of the library's API.

In contrast, while Goedel's output is compilable, it is semantically incorrect. It fails to capture the non-trivial knowledge that one-sided ideals are represented by `Submodule R R` and `Submodule (MulOpposite R) R`, and instead formalizes all ideals as `Ideal R`, which stands only for two-sided ideals, causing the formalization to deviate entirely from the original mathematical intent.

Aria's formalization is both syntactically and semantically correct. It uses the proper Mathlib types for the different ideals but also achieves good readability and modularity with a prerequisite formal definition for nilpotent ideals (`def IsNil`). This success highlights the core advantage of Aria's GoT architecture. Its ability to perform high-level conceptual planning, while strictly grounding the formalization process in Mathlib, allows it to avoid both the API hallucinations of Gemini and the semantic errors of Goedel.

## A.2 EXAMPLE 2: EXISTENCE OF BALANCED BIG COHEN–MACAULAY MODULES CONJECTURE

INFORMAL STATEMENT

Let $R$ be a Noetherian commutative local ring with maximal ideal $m_R$. There exists a $R$-module $W$ such that $m_R W \neq W$ and every system of parameters for $R$ is a regular sequence on $W$.

ARIA GENERATED STATEMENT

```
import Mathlib

/-- A ring `R` is a Noetherian local ring if it is both Noetherian and
    a local ring. -/
class IsNoetherianLocalRing (R : Type*) [CommRing R]
  extends IsNoetherianRing R, IsLocalRing R : Prop

-- Note: `ringKrullDim` is already defined in Mathlib.

-- Note: `Ideal.IsPrimary` is standard in Mathlib and its definition
-- matches the one provided in the context.
```

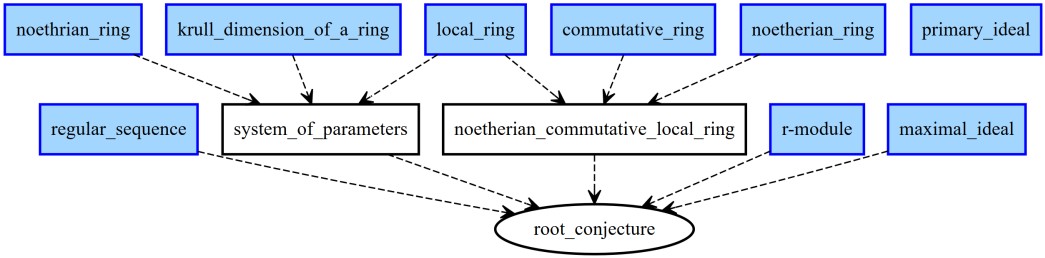

Figure 4: Dependency Graph of Existence of Balanced Big Cohen–Macaulay Modules Conjecture

```
/--
In a Noetherian local ring `R` of Krull dimension `d`, a "system of
    parameters" is a
sequence of `d` elements `x₁, ..., xₙ` in the maximal ideal `m` such
    that the ideal
`I = <x₁, ..., xₙ>` generated by these elements is `m`-primary (i.e., it
    is a primary ideal
whose radical is the maximal ideal `m`).
-/
def IsSystemOfParameters {R : Type*} [CommRing R]
    [IsNoetherianLocalRing R]
    (d : ℕ) (x : Fin d → R) : Prop :=
  ringKrullDim R = ↑d
  (∀ i, x i ∈ LocalRing.maximalIdeal R) ∧
  (let I := Ideal.span (Set.range x);
   I.IsPrimary ∧ I.radical = LocalRing.maximalIdeal R)

/--
Let R be a Noetherian commutative local ring with maximal ideal m_R.
There exists a R-module W such that m_R W ≠ W and every system of
    parameters
for R is a regular sequence on W.
-/
theorem exists_module_where_system_of_parameters_is_regular_sequence
    (R : Type*) [CommRing R] [IsNoetherianLocalRing R] :
    ∃ (W : Type*) (_ : AddCommGroup W) (_ : Module R W),
      (LocalRing.maximalIdeal R) · (⊤ : Submodule R W) ≠ ⊤ ∧
      ∀ (d : ℕ) (x : Fin d → R),
        IsSystemOfParameters d x → RingTheory.Sequence.IsRegular W
    (List.ofFn x) := by
  sorry
```

GOEDEL GENERATED STATEMENT

```
import Mathlib
import Aesop
set_option maxHeartbeats 0
open BigOperators Real Nat Topology Rat
theorem statement (R : Type*) [CommRing R] [IsLocalRing R]
    [IsNoetherianRing R] :
  ∃ (W : Type*) [AddCommGroup W] [Module R W],
      (maximalIdeal R : Ideal R) · W ≠ W ∧
          ∀ (s : Set R), s.Finite ∧ Ideal.span s = maximalIdeal R →
                ∀ (f : R → W), (∀ r ∈ s, f r ≠ 0) → LinearIndependent
    R f := by sorry
```

GEMINI GENERATED STATEMENT

```
import Mathlib

open LocalRing

-- The proposition is defined within a `variable` block,
-- which sets the context for the ring R.
variable (R : Type*) [CommRing R] [IsNoetherianRing R] [LocalRing R]

/--
Formalization of the statement:
"There exists an R-module W such that m_R * W ≠ W and every system
 of parameters for R is a regular sequence on W."
-/
def module_with_regular_sop_exists : Prop :=
  -- "There exists an R-module W..."
  ∃ (W : Type*) [AddCommGroup W] [Module R W],
    -- "...such that m_R * W ≠ W..."
    (maximalIdeal R) · (⊤ : Submodule R W) ≠ \top ∧
    -- "...and every system of parameters for R..."
    ∀ {s : Fin (Ring.Dimension R) → (maximalIdeal R)},
      (IsSystemOfParameters s) →
      -- "...is a regular sequence on W."
      (IsRegularSequence (fun i ↦ (s i : R)) W)
```

### ANALYSIS

This is a more complex conjecture, and the three generated examples once again highlight the poor performance of a specialized model like Goedel in capturing complex mathematical semantics, as well as the knowledge gaps of a general reasoning model like Gemini regarding the Mathlib library.

In this instance, Gemini uses a definition `IsSystemOfParameters` which does not exist in the library.   And for the concept of a regular sequence, which is available in Mathlib (`RingTheory.Sequence.Isregular`), it used an incorrect interface (`IsRegularSequence`).  Goedel, on the other hand, failed entirely to grasp the concepts of "regular sequence" and "system of parameters", producing a completely erroneous statement.

In contrast, Aria's Graph-of-Thought (GoT) architecture helped it correctly identify the key concepts. It successfully searched within Mathlib to find `RingTheory.Sequence.IsRegular`. By leveraging the deconstruct capabilities of GoT, Aria successfully formalize the definition for system of parameters, and ultimately succeeded in formalizing the entire conjecture.

## B  DISCUSSION AND CASE STUDY FOR ARIASCORER

By grounding evaluation in retrieved Lean term information, our checker captures the precise semantics of formal statements with greater accuracy, improving both matching and evaluation quality. In the following section, we present examples showing how the three key innovations contribute to the enhanced performance of AriaScorer.

### B.1  IMPLICIT SEMANTIC INCLUSION

#### INFORMAL STATEMENT

$\mathbb{C}[x, y, z]/(x^2 + y^3 + z^7)$ is a UFD.

#### CONDITIONS AND CONCLUSIONS

**Condition:**
    1. Let R be the ring $R = \mathbb{C}[x, y, z]/(x^2 + y^3 + z^7)$.
**Conclusion:**

- R is a Unique Factorization Domain (UFD). This means:

  1. R is an integral domain (a commutative ring with $1 \neq 0$ and no zero divisors).
  2. For every non-zero, non-unit element $a \in R$, there exist irreducible elements $p_1, p_2, ..., p_k \in R$ such that $a = p_1 p_2 \cdots p_k$.
  3. This factorization is unique: if $a = q_1 q_2 \cdots q_m$ is another factorization into irreducibles, then $k = m$ and there exists a permutation $\sigma$ of $\{1, 2, ..., k\}$ and units $u_1, u_2, ..., u_k \in R$ such that $p_i = u_i q_{\sigma(i)}$ for all $i \in \{1, 2, ..., k\}$.

FORMAL STATEMENT TRANSLATED BY ARIA

```
import Mathlib

-- To state that the quotient ring is a UFD, Lean must first know it's
   an integral domain.
-- This requires the ideal to be prime. The error 'unknown constant'
   indicates that the
-- function name `Ideal.isPrime_iff_isDomain_quotient` used in the
   previous attempt does not exist.
-- The correct mathlib theorem is `Ideal.Quotient.isDomain_iff_prime`,
   which states
-- `IsDomain (R / I) ↔ I.IsPrime`. To prove the `IsDomain` instance
   from the primality of the
-- ideal, we need the reverse direction of this equivalence, which is
   accessed with `.mpr`.

-- The `noncomputable` keyword is required because types like
   `MvPolynomial` over `ℂ`
-- are defined using classical logic and do not have an executable
   implementation.
noncomputable def kleinianSingularityIdeal : Ideal (MvPolynomial (Fin
   3) ℂ) :=
  Ideal.span {
    (MvPolynomial.X 0 : MvPolynomial (Fin 3) ℂ)^2 +
    (MvPolynomial.X 1)^3 +
    (MvPolynomial.X 2)^7
  }

-- We define an abbreviation for the quotient ring type for clarity.
abbrev CxyzModKleinian := MvPolynomial (Fin 3) ℂ /
   kleinianSingularityIdeal

-- We manually provide the `IsDomain` instance for our quotient ring.
   Its proof follows
-- from the fact that the ideal is prime, a non-trivial result which we
   assume with `sorry`.
local instance : IsDomain CxyzModKleinian :=
  (Ideal.Quotient.isDomain_iff_prime kleinianSingularityIdeal).mpr sorry

theorem isUFD_of_C_xyz_mod_x2_y3_z7 :
  UniqueFactorizationMonoid CxyzModKleinian := by
  sorry
```

RETRIEVED LEAN TERMS

```
{'name': ['MvPolynomial'], 'kind': 'definition', 'value': 'Type u_1 →
   (R : Type u_2) → [inst : CommSemiring R] → Type (max u_1 u_2)',
   'informal_name': 'Multivariate polynomials over R with variables in
   σ', 'informal_description': 'The type of multivariate polynomials
   over a commutative semiring R with variables indexed by a type σ.
   Each polynomial is a finite formal linear combination of monomials,
   where a monomial is a product of variables X_i for i ∈ σ raised to
   non-negative integer powers, with coefficients in R.'}
```

```
{'name': ['Ideal', 'span'], 'kind': 'definition', 'value': '{α : Type
    u} → [inst : Semiring α] → Set α → Ideal α', 'informal_name':
    'Ideal generated by a subset', 'informal_description': 'Given a
    subset s of a semiring α, the ideal span(s) is the smallest ideal of
    α containing s, defined as the intersection of all ideals
    containing s.'}
```

MATCHING AND ASSESSMENT BY ARIASCORER

LLM Full Output: Let's compare the mathematical conditions and conclusions with the Lean 4 formal statement one by one:

1. **Let R be the ring** $R = \mathbb{C}[x, y, z]/(x^2 + y^3 + z^7)$:

   - Math: The object of study is a quotient ring. The base ring is the ring of polynomials in three variables $x, y, z$ with complex coefficients, $\mathbb{C}[x, y, z]$. The ideal is the principal ideal generated by the polynomial $x^2 + y^3 + z^7$.
   - Lean: `abbrev CxyzModKleinian := MvPolynomial (Fin 3) ℂ ⁄ kleinianSingularityIdeal`. This defines the ring as a quotient $⁄$. The base ring is `MvPolynomial (Fin 3) ℂ`, which represents multivariate polynomials with coefficients in ℂ (complex numbers) and variables indexed by `Fin 3` (a type with 3 elements, corresponding to $x, y, z$). The ideal `kleinianSingularityIdeal` is defined as `Ideal.span` of the set containing the polynomial `(MvPolynomial.X 0)^2 + (MvPolynomial.X 1)^3 + (MvPolynomial.X 2)^7`, which corresponds exactly to $x^2 + y^3 + z^7$.
   - Match: Perfectly match.

2. **R is a Unique Factorization Domain (UFD)**:

   - Math: The ring R is an integral domain, and every non-zero, non-unit element has a unique factorization into irreducible elements (up to order and multiplication by units).
   - Lean: The conclusion is formalized in two parts. First, `local instance : IsDomain CxyzModKleinian` asserts that the ring is an integral domain. Second, the main theorem `theorem isUFD_of_C_xyz_mod_x2_y3_z7 : UniqueFactorizationMonoid CxyzModKleinian` asserts the property of unique factorization. In Mathlib, a ring is a UFD if and only if it satisfies both `IsDomain` and `UniqueFactorizationMonoid`.
   - Match: Perfectly match.

**Check for missing conditions / implicit conditions:**

   - The mathematical statement "R is a UFD" contains the implicit condition that R must be an integral domain. The Lean code makes this explicit with the `local instance : IsDomain CxyzModKleinian`. This is not a missing condition but a correct formalization of an implicit part of the mathematical definition.
   - Match: Perfectly match.

ANALYSIS

In this case, the informal statement asserts that the quotient ring $R = \mathbb{C}[x, y, z]/(x^2 + y^3 + z^7)$ is a Unique Factorization Domain (UFD), which requires $R$ to be an integral domain by definition. AriaScorer, empowered by term-level retrieval, correctly accesses the definitions of relevant Lean terms and tracks newly written local instances. This allows it to recognize that the concept of UFD is semantically realized through the combination of `IsDomain` and `UniqueFactorizationMonoid`, even though they appear separately in the formal code.

Without this term-level semantic grounding, a checker may mistakenly treat the newly defined local instance as a nontrivial addition to the conclusion, rather than recognizing it as a prerequisite implicitly embedded in the definition of `UniqueFactorizationMonoid`. By contrast, AriaScorer uncovers this internal dependency and correctly matches the formalization. This case illustrates the strength of semantic grounding: it enables the checker to move beyond surface forms and reason

about the structural meaning and interrelation of formal terms—something that purely textual comparison would fail to capture.

## B.2 DEFINITION DISCREPANCY DETECTION

### INFORMAL STATEMENT

Let $A, B \in \mathbb{Q}^{\times}$ be rational numbers. Consider the quaternion ring $D_{A,B,\mathbb{R}} = \{a + b\boldsymbol{i} + c\boldsymbol{j} + d\boldsymbol{k} \mid a, b, c, d \in \mathbb{R}\}$ in which the multiplication satisfies relations: $\boldsymbol{i}^2 = A$, $\boldsymbol{j}^2 = B$, and $\boldsymbol{ij} = -\boldsymbol{ji} = \boldsymbol{k}$. Show that $D_{A,B,\mathbb{R}}$ is either isomorphic to $\mathbb{H}$ (Hamilton quaternion) or isomorphic to $\mathrm{Mat}_{2\times 2}(\mathbb{R})$ as $\mathbb{R}$-algebras.

### CONDITIONS AND CONCLUSIONS

**Conditions:**

1. $A, B \in \mathbb{Q}^{\times}$ (i.e., $A$ and $B$ are non-zero rational numbers).
2. $D_{A,B,\mathbb{R}}$ is an $\mathbb{R}$-algebra defined on the set $\{a + b\boldsymbol{i} + c\boldsymbol{j} + d\boldsymbol{k} \mid a, b, c, d \in \mathbb{R}\}$.
3. The multiplication in $D_{A,B,\mathbb{R}}$ is defined by the relations:

   - $\boldsymbol{i}^2 = A$
   - $\boldsymbol{j}^2 = B$
   - $\boldsymbol{ij} = -\boldsymbol{ji} = \boldsymbol{k}$

**Conclusion:**

- $D_{A,B,\mathbb{R}}$ is isomorphic to $\mathbb{H}$ (the Hamilton quaternions) or isomorphic to $\mathrm{Mat}_{2\times 2}(\mathbb{R})$ (the algebra of $2 \times 2$ real matrices) as $\mathbb{R}$-algebras. This can be stated as: $(D_{A,B,\mathbb{R}} \cong \mathbb{H}) \vee (D_{A,B,\mathbb{R}} \cong \mathrm{Mat}_{2\times 2}(\mathbb{R}))$, where $\cong$ denotes an isomorphism of $\mathbb{R}$-algebras.

### FORMAL STATEMENT TRANSLATED BY ARIA

```
import Mathlib

theorem
    quaternion_algebra_over_reals_isomorphic_to_hamiltonians_or_matrices
    (A B : ℚ) (hA : A ≠ 0) (hB : B ≠ 0) :
    Nonempty (QuaternionAlgebra ℝ (A : ℝ) (B : ℝ) 1 ≅ₐ[ℝ]
    (QuaternionAlgebra ℝ (-1) (-1) 1)) ∨
    Nonempty (QuaternionAlgebra ℝ (A : ℝ) (B : ℝ) 1 ≅ₐ[ℝ] Matrix (Fin
    2) (Fin 2) ℝ) := by
  sorry
```

### RETRIEVED LEAN TERMS

```
{'name': ['Matrix'], 'kind': 'definition', 'value': "Type u → Type u'
    → Type v → Type (max u u' v)", 'informal_name': 'Matrix',
    'informal_description': 'The type of matrices with rows indexed by
    a type m, columns indexed by a type n, and entries in a type α. A
    matrix M can be represented as a function M : m → n → α, where M(i,j)
    gives the entry in row i and column j.'}
{'name': ['QuaternionAlgebra'], 'kind': 'structure', 'value': '(R :
    Type u_1) → R → R → R → Type u_1', 'informal_name': 'Quaternion
    Algebra', 'informal_description': 'The structure representing a
    quaternion algebra over a commutative ring R with fixed
    coefficients a, b, c, where the elements are of the form x + yi + zj + wk
    with x, y, z, w ∈ R and the multiplication rules are determined by
    i² = a + bi and j² = c. The structure has four components: the real part
    `re` and three imaginary parts `imI`, `imJ`, `imK`.'}
```

### MATCHING AND ASSESSMENT BY ARIA'S SEMANCTIC CHECKER

Let's compare the mathematical conditions and conclusions with the Lean 4 formal statement one by one:

1. $A, B \in \mathbb{Q}^{\times}$ **(i.e., $A$ and $B$ are non-zero rational numbers).**:

- Math: $A, B \in \mathbb{Q}^{\times}$.
- Lean: (A B : $\mathbb{Q}$) (hA : A $\neq$ 0) (hB : B $\neq$ 0).
- Match: Perfectly match.

2. $D_{A,B,\mathbb{R}}$ **is an $\mathbb{R}$-algebra defined on the set** $\{a + bi + cj + dk \mid a, b, c, d \in \mathbb{R}\}$**.**:

- Math: The algebra is defined on the set of elements of the form $a + bi + cj + dk$ with real coefficients.
- Lean: QuaternionAlgebra $\mathbb{R}$ (A : $\mathbb{R}$) (B : $\mathbb{R}$) 1.
- According to the provided Mathlib entry for QuaternionAlgebra, the elements are of the form x + yi + zj + wk where x, y, z, w are from the base ring R, which is $\mathbb{R}$ in this case. This matches the form of the elements.
- Match: Perfectly match.

3. **The multiplication in $D_{A,B,\mathbb{R}}$ is defined by the relations: $i^2 = A$, $j^2 = B$, and $ij = -ji = k$.**:

- Math: The multiplication rules are $i^2 = A$ and $j^2 = B$.
- Lean: The multiplication is determined by QuaternionAlgebra $\mathbb{R}$ (A : $\mathbb{R}$) (B : $\mathbb{R}$) 1.
- According to the provided Mathlib entry, QuaternionAlgebra R a b c has multiplication rules i^2 = a + bi and j^2 = c. For the Lean statement, R is $\mathbb{R}$, a is (A : $\mathbb{R}$), b is (B : $\mathbb{R}$), and c is 1. This implies the rules are $i^2 = A + Bi$ and $j^2 = 1$. These rules do not match the mathematical rules.
- Match: Major inconsistency.

4. $D_{A,B,\mathbb{R}}$ **is isomorphic to $\mathbb{H}$ (the Hamilton quaternions) or isomorphic to $\mathrm{Mat}_{2\times2}(\mathbb{R})$ (the algebra of 2x2 real matrices) as $\mathbb{R}$-algebras.**:

- Math: $(D_{A,B,\mathbb{R}} \cong \mathbb{H}) \vee (D_{A,B,\mathbb{R}} \cong \mathrm{Mat}_{2\times2}(\mathbb{R}))$.
- Lean: Nonempty (QuaternionAlgebra $\mathbb{R}$ (A : $\mathbb{R}$) (B : $\mathbb{R}$) 1 $\simeq_a[\mathbb{R}]$ (QuaternionAlgebra $\mathbb{R}$ (-1) (-1) 1)) $\vee$ Nonempty (QuaternionAlgebra $\mathbb{R}$ (A : $\mathbb{R}$) (B : $\mathbb{R}$) 1 $\simeq_a[\mathbb{R}]$ Matrix (Fin 2) (Fin 2) $\mathbb{R}$).
- The conclusion in Lean concerns the algebra QuaternionAlgebra $\mathbb{R}$ (A : $\mathbb{R}$) (B : $\mathbb{R}$) 1, which, as established in point 3, is not the same as $D_{A,B,\mathbb{R}}$. Furthermore, the Hamilton quaternions ($\mathbb{H}$), defined by $i^2 = -1, j^2 = -1$, are represented in Lean as QuaternionAlgebra $\mathbb{R}$ (-1) (-1) 1. According to the provided documentation, this corresponds to an algebra with rules $i^2 = -1 - i$ and $j^2 = 1$, which is not $\mathbb{H}$. While the representation of $\mathrm{Mat}_{2\times2}(\mathbb{R})$ as Matrix (Fin 2) (Fin 2) $\mathbb{R}$ is correct, the overall statement is about different algebraic structures than those in the mathematical problem.
- Match: Major inconsistency.

**Check for missing conditions / implicit conditions:**

- No missing conditions / implicit conditions
- Match: Perfectly match.

ANALYSIS

In this case, the informal statement introduces $D_{A,B,\mathbb{R}}$ with multiplication rules: $i^2 = A$, $j^2 = B$, and $ij = -ji = k$. However, the corresponding Lean term actually has completely different rules: $i^2 = A + Bi, j^2 = 1$. These differences are not perceptible by surface text comparison, Both the informal and formal version use $A$ and $B$, and have similar algebra structure. By term-level grounding, AriaChecker gets the explicit retrieval of the definition of QuaternionAlgebra, making the semantic incorrectness obvious to AriaScorer. While LeanScorer is deceived by apparent textual similarity and misjudges this case, AriaScorer correctly flags it as "Major Inconsistency". This demonstrates that Defintion Discrepancy Detection enables AriaScorer to distinguish semantic

incorrectness, ensuring that the formal statement aligns with the formal statement in mathematical intent.

### B.3 HALLUCINATION SUPPRESSION VIA GROUNDING

INFORMAL STATEMENT

Let $\alpha = \sqrt{(2+\sqrt{2})(3+\sqrt{3})}$ and consider the extension $E = \mathbb{Q}(\alpha)$. Show that $\mathrm{Gal}(E/\mathbb{Q}) \cong Q_8$, the quaternion group of order 8.

CONDITIONS AND CONCLUSIONS

Conditions:

1. $\alpha = \sqrt{(2+\sqrt{2})(3+\sqrt{3})}$
2. $E = \mathbb{Q}(\alpha)$ (E is the smallest field containing both $\mathbb{Q}$ and $\alpha$)

Conclusion:

- The Galois group of the extension $E$ over $\mathbb{Q}$ is isomorphic to the quaternion group $Q_8$. In mathematical notation: $\mathrm{Gal}(E/\mathbb{Q}) \cong Q_8$, where:
  - $\mathrm{Gal}(E/\mathbb{Q}) = \{\sigma : E \to E \mid \sigma$ is a field automorphism and $\forall x \in \mathbb{Q}, \sigma(x) = x\}$
  - $Q_8 = \langle x, y \mid x^4 = 1, x^2 = y^2, y^{-1}xy = x^{-1} \rangle$ is the quaternion group of order 8.

FORMAL STATEMENT TRANSLATED BY ARIA

```
import Mathlib

open Polynomial

/--
Let α = √(2+√2)(3+√3). The field E = ℚ(α) is a Galois
extension of ℚ, and its Galois group is isomorphic to the quaternion
    group Q₈.
...     Nonempty (P.Gal ≅* QuaternionGroup 1) :=
  sorry
```

RETRIEVED LEAN TERMS

```
{'name': ['IntermediateField', 'adjoin'], 'kind': 'definition',
    'value': '(F : Type u_1) →\n  [inst : Field F] → {E : Type u_2} →
    [inst_1 : Field E] → [inst_2 : Algebra F E] → Set E →
    IntermediateField F E', 'informal_name': 'Field adjunction of a set
    S to F', 'informal_description': 'Given a field extension E of F
    and a subset S ⊆ E, the structure `IntermediateField.adjoin F S`
    represents the intermediate field obtained by adjoining the
    elements hedral group; when n = 1, it is isomorphic to a cyclic
    group of order 4.'}
{'name': ['Polynomial'], 'kind': 'structure', 'value': '(R : Type u_1)
    → [inst : Semiring R] → Type u_1', 'informal_name': 'Univariate
    polynomials over a semiring', 'informal_description': 'The
    structure `Polynomial R` represents the type of univariate
    polynomials over a semiring `R`, denoted as `R[X]`. Polynomials are
    constructed using the variable `X` and constant embedding `C` from
    `R`, forming a semiring structure.'}
```

MATCHING AND ASSESSMENT BY ARIA'S SEMANCTIC CHECKER

Let's compare the mathematical conditions and conclusions with the Lean 4 formal statement one by one:

1. **Condition:** $\alpha = \sqrt{((2 + \sqrt{2})(3 + \sqrt{3}))}$:

   - Math: $\alpha$ is defined constructively by the given formula, which specifies a particular (positive real) number.
   - Lean: `let P : ℚ[X] := X^8 - 24 * X^6 + 144 * X^4 - 288 * X^2 + 144` and $\exists\ \alpha \in$ `rootSet P P.SplittingField`, `....` This does not define $\alpha$ using the `sqrt` formula. Instead, it posits the existence of an element $\alpha$ that is a root of a specific polynomial `P`. While `P` is the minimal polynomial of the mathematical $\alpha$, the formal statement is about *any* root of `P`, not the specific one in the math problem.
   - Match: Minor Inconsistency.

2. **Condition:** $E = \mathbb{Q}(\alpha)$ (**E is the smallest field containing both $\mathbb{Q}$ and $\alpha$**):

   - Math: $E$ is defined as the field extension $\mathbb{Q}(\alpha)$. The problem then asks to compute the Galois group of this extension, which implicitly assumes or requires that $E/\mathbb{Q}$ is a Galois extension.
   - Lean: `IntermediateField.adjoin ℚ {α} = ⊤.` The term `IntermediateField.adjoin ℚ {α}` corresponds to $\mathbb{Q}(\alpha)$. The term $\top$ (top) refers to the ambient field, which is `P.SplittingField`. Thus, the statement asserts that $\mathbb{Q}(\alpha)$ is the splitting field of `P`. This makes the implicit mathematical assumption that `E/ℚ` is Galois into an explicit and strong condition within the theorem.
   - Match: Minor Inconsistency.

2. **Conclusion:** $\mathrm{Gal}(E/\mathbb{Q}) \cong Q_8$:

   - Math: The Galois group of the extension $E$ over $\mathbb{Q}$ is isomorphic to $Q_8$, the quaternion group of order 8.
   - Lean: `Nonempty (P.Gal ≃* QuaternionGroup 1).` `P.Gal` corresponds to $\mathrm{Gal}(E/\mathbb{Q})$. `Nonempty (G ≃* H)` means G is isomorphic to H. However, `QuaternionGroup 1` is the generalized quaternion group with `n=1`, which is isomorphic to the cyclic group of order 4 (`C_4`). The quaternion group of order 8, `Q_8`, corresponds to `QuaternionGroup 2`.
   - Match: Major Inconsistency.

**Check for missing conditions / implicit conditions:**

   - The Lean statement introduces the explicit definition of the polynomial `P`, which is not present in the original mathematical problem statement but is a necessary intermediate result for its proof. This is an added condition. No conditions are missing.
   - Match: Minor Inconsistency.

MISJUDGMENT WHEN TERM-LEVEL GROUNDING IS ELIMINATED

Let's compare the mathematical conditions and conclusions with the Lean 4 formal statement one by one:

1. **Definition of $\alpha$:**

   - Math: $\alpha = \sqrt{(2 + \sqrt{2})(3 + \sqrt{3})}$. This defines $\alpha$ as a specific positive real number.
   - Lean: `let P : ℚ[X] := X^8 - 24 * X^6 + 144 * X^4 - 288 * X^2 + 144` and $\exists\ \alpha \in$ `rootSet P P.SplittingField`, `....` This implicitly defines $\alpha$ as an arbitrary root of the given polynomial `P`

(which is the minimal polynomial of the mathematical $\alpha$). This is a shift from a concrete definition to an abstract algebraic one.
- Match: Minor Inconsistency.

2. **Definition of the field E**:

- Math: $E = \mathbb{Q}(\alpha)$. The field is defined by adjoining the specific number $\alpha$ to $\mathbb{Q}$. That this field is a Galois extension over $\mathbb{Q}$ is a substantial part of the proof.
- Lean: The ambient field is `P.SplittingField`, the splitting field of the polynomial P. The condition `IntermediateField.adjoin` $\mathbb{Q}$ `{`$\alpha$`}` `=` $\top$ states that this splitting field is generated by a single root $\alpha$. This reframes the definition of the field: the problem is set in the splitting field of P, and one of the goals is to show it can be generated by a single root.
- Match: Minor Inconsistency.

3. **The groups being compared**:

- Math: $\mathrm{Gal}(E/\mathbb{Q})$ (the Galois group of the extension $E$ over $\mathbb{Q}$) and $Q_8$ (the quaternion group).
- Lean: `P.Gal` and `QuaternionGroup 1`. `P.Gal` is the Galois group of `P.SplittingField` over $\mathbb{Q}$, which corresponds to $\mathrm{Gal}(E/\mathbb{Q})$. The docstring and context imply `QuaternionGroup 1` is the name used for $Q_8$.
- Match: Perfectly match.

4. **Conclusion**:

- Math: $\mathrm{Gal}(E/\mathbb{Q}) \cong Q_8$, i.e., the Galois group is isomorphic to the quaternion group.
- Lean: `Nonempty (P.Gal ≃* QuaternionGroup 1)`. This states that there exists a group isomorphism between `P.Gal` and `QuaternionGroup 1`. The `Nonempty` wrapper is a standard way in Lean to express existence in a proposition.
- Match: Perfectly match.

**Check for missing conditions / implicit conditions:**

- The mathematical problem requires implicitly that one finds the minimal polynomial of $\alpha$ and proves that the extension $\mathbb{Q}(\alpha)/\mathbb{Q}$ is Galois (i.e., is the splitting field of this polynomial). The Lean statement makes these aspects explicit by providing the polynomial P from the start and including the condition `IntermediateField.adjoin` $\mathbb{Q}$ `{`$\alpha$`}` `=` $\top$ (that $\mathbb{Q}(\alpha)$ is the splitting field) as part of the theorem to be proven. The formal statement is more explicit, which is a feature of formalization, not a missing condition.
- Match: Perfectly match.

ANALYSIS

In this case, the Lean statement claims an isomorphism `Nonempty (P.Gal ≃* QuaternionGroup 1)`, using the key Lean term `QuaternionGroup 1`. It is easily to assume that `QuaternionGroup 1` refers to the quaternion group of order 8, $Q_8$. Actually, `QuaternionGroup 1` is isomorphic to the cyclic group $C_4$, while the actual representation of $Q_8$ is `QuaternionGroup 2`. This subtle but important distinction is overlooked when LLM gives the checking purely on surface texts. In the setting without the information of Lean terms, checker is misled by the hallucination of LLM and gives a wrong judgment. In comparison, AriaScorer grounds the checking pipeline in concrete semantics. With the usage of the definition of `QuaternionGroup n`, AriaScorer correctly flags the statement as "Major Inconsistency", which is in line with human annotation. It gives an example of how the process of hallucination suppression constrains the LLM's reasoning within Lean terms, guarantees precision in the semantic correctness checking.

## B.4 VERIFICATION STRATEGY AND ERROR PROPAGATION ANALYSIS

### B.4.1 VERIFICATION STRATEGY

A critical architectural decision in our system is the application of AriaScorer exclusively as a terminal evaluator rather than an iterative feedback signal. This design is driven by a strategic trade-off between verification rigor, algorithmic stability, and computational efficiency. AriaScorer is engineered to function as a rigorous, independent checker involving multiple LLM calls and database searches. Consequently, it is computationally expensive and best suited for validating the quality of the completed solution.

Our preliminary experiments with iterative semantic feedback revealed two primary challenges:

- **Instability:** Correcting semantic issues at intermediate steps frequently disrupted the syntactic structure of the proof, leading to oscillatory behavior where the system toggled between semantic and syntactic errors.
- **Inference Efficiency:** Given the computational intensity of AriaScorer, applying it to every intermediate node would drastically increase the total inference time. This inefficiency is further compounded by the aforementioned oscillatory behavior.

Furthermore, utilizing AriaScorer as a feedback signal for reflection-in addition to its role as the final checker-introduces a risk of self-referential bias, where the formalizer might learn to overfit the scorer's specific preferences rather than producing universally correct proofs.

### B.4.2 ANALYSIS OF SEMANTIC ERROR PROPAGATION

We specifically investigated the risk of semantic error propagation, where an incorrect intermediate definition might lead to a finalized but flawed proof. Our empirical data suggests that errors caused by "correctly typed but semantically wrong" intermediate definitions are statistically rare. A root cause analysis of failure cases in the FATE-X dataset revealed that only a single instance of failure was attributable to a flaw in definition synthesis that propagated to cause a final semantic inconsistency.

**Case Study: The Catenary Ring**   To demonstrate the rigor of AriaScorer in detecting such rare propagation cases, we present a detailed analysis of the "Catenary Ring" instance. In this case, the generator synthesized a definition for `CatenaryRing` that was syntactically valid but mathematically overly restrictive compared to the standard definition.

```
import Mathlib

universe u

/--
A ring `R` is a catenary ring if it is a commutative Noetherian ring,
    and for any two
prime ideals `p` and `q` with `p ⊂ q`, all saturated chains of prime
    ideals between
`p` and `q` have the same length.

The length of a finite chain of prime ideals `p_0 ⊂ p_1 ⊂ ... ⊂ p_n` is
    `n`. This corresponds
to the chain having `n+1` elements. Thus, two chains have the same
    length if and only
if they have the same number of elements (cardinality).
-/
class CatenaryRing (R : Type u) [CommRing R] extends IsNoetherianRing R
    : Prop where
  /--
  For any two prime ideals `p, q` with `p ≤ q`, all saturated (i.e.
    maximal) chains
  of prime ideals in the interval `Set.Icc p q` have the same
    cardinality.
```

```
  -/
  eq_saturated_chain_length :
    ∀ (p q : PrimeSpectrum R), p ≤ q →
      ∀ (c₁ c₂ : Set (Set.Icc p q)),
        IsMaxChain (· ≤ ·) c₁ →
        IsMaxChain (· ≤ ·) c₂ →
        Set.ncard c₁ = Set.ncard c₂

/--
A Noetherian Unique Factorization Domain of Krull dimension at most 3
   is catenary.
-/
theorem ufd_of_dim_le_three_is_catenary (A : Type u)
    [CommRing A] [IsDomain A] [IsNoetherianRing A]
    [UniqueFactorizationMonoid A]
    (h_dim : Order.krullDim (PrimeSpectrum A) ≤ 3) :
    CatenaryRing A :=
  sorry
```

Below is the analysis of AriaScorer:

```
**A is a catenary ring**:
- Math: A is a catenary ring, i.e., for any two prime ideals P,Q of A
    with P ⊆ Q, any two saturated chains of prime ideals between P and
    Q have the same length.
- Lean: `CatenaryRing A`. The provided `class CatenaryRing` definition
    `extends IsNoetherianRing R`, which means that to be a
    `CatenaryRing`, a ring must be Noetherian in addition to satisfying
    the chain condition. The mathematical definition provided does not
    require the ring to be Noetherian. This makes the Lean conclusion
    stronger than the mathematical one.
- Match: Major inconsistency.
```

Even though the specific problem context (Theorem `ufd_of_dim_le_three_is_catenary`) explicitly included the Noetherian ring condition (`[IsNoetherianRing A]`), AriaScorer correctly identified that the definition itself hallucinated an unnecessary inheritance (`extends IsNoetherianRing`). This demonstrates that AriaScorer maintains rigorous judgment even when the propagated error is subtle and contextually masked.

### B.4.3 FUTURE OUTLOOK

While the current dependency graphs in FATE-X are relatively shallow (averaging 2-3 layers), minimizing the impact of error propagation, this challenge may become more pronounced in large-scale formalization tasks, such as formalizing entire textbooks. Future work may aim to efficiently integrate semantic signals into larger systems by adopting strategic checkpointing (e.g., verifying every $k$ layers) to balance efficiency and correctness.

### B.5 LIMITATIONS OF SYNTACTIC METRICS IN RESEARCH-LEVEL MATHEMATICS

A potential concern in using a LLM-based evaluator is the risk of self-referential bias. We address this by adhering to a strict architectural decoupling: AriaScorer is utilized solely as a post-hoc evaluator and is never exposed to the agent during generation or reflection. This ensures that the observed performance gains reflect genuine capabilities rather than optimization towards the metric. Furthermore, the reliability of AriaScorer has been validated against human expert annotations on the FATE-X and Conjecture datasets, achieving a 95.5% alignment rate.

#### B.5.1 WHY SYNTACTIC METRICS ARE LIMITED.

While syntactic metrics such as BEq or simple type-checking are standard for simple formalization tasks, we find them ill-suited for research-level mathematics. At this level, proving logical equivalence between a synthesized definition and a reference statement is often non-trivial. Stan-

dard syntactic matchers frequently generate false negatives due to several intrinsic complexities of formal libraries:

1. **Multiple Mathematical Definitions:** A single concept often has multiple equivalent mathematical definitions. Different contexts (or authors) may prefer different formulations, leading to distinct structures or type classes in Lean.

2. **Bundled Type Classes:** Structures can be "bundled" differently. For example, a two-variable polynomial ring can be formalized as `MvPolynomial (Fin 2) R` or `Polynomial (Polynomial R)`. These are not definitionally equal; proving their equivalence requires constructing a complex algebraic isomorphism ($\simeq_a$ `[R]`).

3. **Inheritance Structures (Diamonds):** Type classes inherit from others. Different inheritance paths can lead to "diamond" problems where the formal representations diverge despite representing the same object. This issue is acute in our framework: the Graph of Thoughts (GoT) planner synthesizes deeply structured, multi-layer definition chains. This structured approach naturally induces diamond patterns.

### B.5.2 CASE STUDY: THE E8 KLEINIAN SINGULARITY.

To illustrate why syntactic metrics fail in this domain, we present a specific case from the FATE-X dataset regarding the E8 Kleinian Singularity.

```
--Aria Generated Code:
import Mathlib
noncomputable def kleinian_singularity_E8_polynomial : MvPolynomial (Fin
    3) ℂ :=
  (MvPolynomial.X 0) ^ 2 + (MvPolynomial.X 1) ^ 3 + (MvPolynomial.X 2) ^
    7
abbrev E8_singularity_quotient_ring :=
  (MvPolynomial (Fin 3) ℂ) ╱ (Ideal.span
    {kleinian_singularity_E8_polynomial})
instance kleinian_singularity_E8_ideal_isPrime :
  (Ideal.span {kleinian_singularity_E8_polynomial}).IsPrime := by sorry
theorem isUFD_E8_singularity_quotient_ring :
  UniqueFactorizationMonoid E8_singularity_quotient_ring := by sorry

--Reference Code:
import Mathlib
/--
The ring R = ℂ[x, y, z]/(x² + y³ + z⁷).
-/
abbrev R : Type := (MvPolynomial (Fin 3) ) ╱ Ideal.span {(.X 0 ^ 2 + .X
    1 ^ 3 + .X 2 ^ 7 : MvPolynomial (Fin 3) )}
/--
ℂ[x, y, z]/(x² + y³ + z⁷) is a UFD.
-/
theorem quotient_not_UFD :
  ∃ (h : IsDomain R),
    (UniqueFactorizationMonoid R) := by sorry
```

Both theorems assert the same fact: the coordinate ring of the E8 singularity is a Unique Factorization Domain (UFD).

To formally prove that these two statements are equivalent in Lean (and thus satisfy a strict checker, e.g. BEq), one would need to perform three non-trivial steps:

- Ring Identification: Prove `R = E8_singularity_quotient_ring`. This requires unfolding the definitions, returning to the quotient construction, and applying rewrite tactics.

- Domain Verification: Prove that R is an integral domain. This follows from the `kleinian_singularity_E8_ideal_isPrime` instance, but requires a non-trivial proof step.

- Logical Elimination: Prove the lemma (∃ (h : IsDomain R), UniqueFactorizationMonoid R) UniqueFactorizationMonoid R. This involves existential elimination logic.

Bridging this gap requires unfolding 4-5 technical layers and employing intermediate-level Lean tactics. BEq, which operates on structural equality, cannot perform this semantic reasoning. Consequently, we maintain that AriaScorer represents a necessary, reliable, and unbiased standard for this domain.

## C    ABLATION STUDIES

To quantify the individual contributions of the core components within our Aria agent, we conducted a series of comprehensive ablation studies. We systematically disabled the Reflection, Graph-of-Thought (GoT), and Retrieval-Augmented Generation (RAG) modules to measure their impact on the performance. All experiments were conducted on the challenging benchmarks FATE-X and homological conjectures, with the results presented in Table 3 and Table 4.

Table 3: Ablation study results on the Conjectures dataset. Performance drops significantly as key components of Aria are removed, highlighting their individual contributions. All values are success rates (%).

| Configuration | Final acc. |
|---|---|
| **Aria (Full System)** | **42.9** |
| Ablations of Aria: | |
| without Reflection | 0 |
| without GoT | 7.1 |
| without RAG | 0 |
| Baseline (Gemini) | 0 |

Table 4: Ablation study results on the FATE-X benchmark. All values are success rates (%).

| Configuration | Compiler | Final acc. |
|---|---|---|
| **Aria (Full System)** | **69.0** | **44.0** |
| Ablations of Aria: | | |
| without Reflection | 19.0 | 14.0 |
| without GoT | **69.0** | 38.0 |
| without RAG | 61.0 | 43.0 |
| Baseline (Gemini) | 27.0 | 21.0 |

### C.1    ABLATING THE REFLECTION MECHANISM

This study is designed to quantify the contribution of our agent's core iterative self-correction mechanism. In the full Aria agent, each generation step (for both prerequisite definitions and the final theorem) is embedded in a refinement loop that allows for 16 reflection attempts. Within this loop, the agent generates a candidate formal definition or statement and receives feedback from the compiler, and uses this feedback to inform the next generation attempt.

For ablation, we disable the refinement loop entirely, restricting the agent to a single generation attempt at each stage.

As shown in Table 4, ablating the reflection module causes the final accuracy on FATE-X to drop from 44% to 14% and the compilation success rate from 69% to 19%, even lower than that of baseline. This dramatic performance decrease is also observed on the Conjectures dataset, where the success rate plummet from 42.9% to 0%. The result indicates that a single generation is often insufficient for both capturing the semantic nuances and the syntactic rigor of complex mathematical statements. Therefore, we conclude that the Reflection module is a crucial part in our agent's architecture.

## C.2 Ablating the GoT planner

The experimental setup for this ablation is as follows: first, we extract a flat list of conceptual keywords from the original informal statement. Then, for each concept in this list, we use LeanSearch to retrieve it in Mathlib. In contrast to the full system, this process does not perform any further recursive decomposition, regardless of the search outcome. The agent then directly synthesizes the final formal statement only using the results from this search.

**Quantitative Analysis.** As shown in Table 3, the impact of GoT scales with problem difficulty. On the challenging Conjectures dataset, the full Aria system successfully formalized 6 of the 14 conjectures, whereas the version without GoT only managed 1. Similarly, on the FATE-X benchmark (Table 4), removing GoT causes the final accuracy to drop from 44.0% to 38.0%, although the compilation success rate remains constant at 69.0%.

However, Table 5 reveals a counter-intuitive phenomenon on the simpler FATE-H dataset: the ablated agent achieves a higher compilation success rate (95% vs. 89%) but a significantly lower final accuracy (54% vs. 71%).

**The Trade-off between Syntactic Risk and Semantic Rigor.** We attribute the "high compilation, low accuracy" anomaly on FATE-H to a strategic trade-off. The GoT planner prioritizes semantic explicitness by forcing the generation of all necessary intermediate definitions (e.g., explicitly defining extended fields or tensor products as separate structures). While this ensures semantic consistency, it significantly increases the total attack surface for syntactic errors. For instance, the modular style introduces complexities in global namespace management and Lean's type class resolution-where instances of equivalent but distinct definitions often fail to interoperate without explicit equivalence proofs.

On simpler problems like those in FATE-H, which typically require no prerequisite definitions, this structural overhead yields a net negative impact on compilation. The ablated model, which tends to generate monolithic statement using local `let` bindings, avoids these interface complexities but fails to capture the correct semantics, leading to lower final accuracy.

**Scaling to Complexity** This relationship shifts as problem complexity increases. On FATE-X and Conjectures, the "syntactic cost" of longer code is effectively offset by the "structural benefit" of decomposition. Without GoT, the agent's attempt to formalize novel, high-level concepts monolithically leads to 2 distinct failure modes: synthesis failure (inability to generate complex and definitions) and interface hallucination.

In summary, GoT acts as an indispensable engine for the creative mathematical construction demanded by research-level auto-formalization. Unlike prior works that rely solely on static library retrieval, GoT explicitly leverages the reasoning LLM's natural-language mathematical capability to construct dynamic dependency graphs. This enables a modular formalization style that bridges the gap between the model's internal knowledge and the rigorous requirements of the formal system.

Table 5: Performance comparison between the full Aria agent and its GoT-ablated version on the FATE-H benchmark. All values are success rates (%).

| Configuration | Compiler | Final acc. |
|---|---|---|
| **Aria (Full System)** | 89.0 | **71.0** |
| Aria (without GoT) | **95.0** | 54.0 |

## C.3 Ablating the RAG module

To measure the value of Retrieval-Augmented Generation (RAG), we designed this study to contrast live, tool-based retrieval against reliance on the pretrained, static knowledge of the Large Language Model (LLM).

In the full system, the agent's grounding process is executed by leveraging LeanSearch. The LLM's task is confined to reasoning over this verified set of options. For the ablated version, we disable the

retrieval tool entirely. Instead, the agent directly queries the LLM, to recall the correct formal name for a concept based on its own knowledge.

Our ablation studies, presented in Table 3 and Table 4, reveal the crucial role of the RAG module, particularly as problem complexity increases. While the ablation version resulted in only a moderate drop in final accuracy on FATE-X (from 69% to 61%), its effect on the more challenging Conjectures dataset was absolute, with the success rate collapsing from 42.9% to 0%.

This divergence highlights a key insight into the agent's capabilities. For moderately complex tasks like those in FATE-X, the agent can partially compensate for the lack of retrieval through its powerful self-reflection mechanism. By interpreting the compiler's precise feedback on "unknown identifiers," the agent may iteratively rediscover correct Mathlib definitions. However, this trial-and-error recovery process is insufficient for complex conjectures. The 0% accuracy reveals that without the contextual grounding from RAG, the LLM's inaccurate internal knowledge of Mathlib leads it to hallucinate non-existent definitions and confidently judge them as grounded. This foundational error prevents the generation of compilable code, demonstrating that our RAG module is essential for success on challenging mathematical reasoning tasks.

## D PROMPTS

For clarity and reproducibility, we present the prompt frameworks used by Aria across various stages.

---

**Prompt for Decomposition Phase**

```
You are an expert mathematician and a specialist in formal
mathematics, specifically Lean 4 and its library, mathlib4.  Your
task is to deconstruct a given mathematical concept into its
immediate, foundational prerequisite concepts.
The goal is to produce a list of terms that are themselves canonical,
searchable definitions.  I will provide you with examples of correct
deconstruction before giving you the final task.
--
**Example 1:**
- **Input Concept:** "Finitely Generated Prime Ideal"
- **Correct Output:** "dependencies":  ["finitely generated ideal",
"prime ideal"]
...(few shot examples)...
--
**Now, perform the task for the following concept based on its
name.**
**Concept to deconstruct:** "node.name"
```

---

**Prompt for Grounding Phase**

```
You are a meticulous expert in Lean 4 and `mathlib4`.  Your task is
to act as a "grounding" reasoner for a formalization agent.  Your
goal is to determine if a given mathematical concept has a canonical
formal definition in `mathlib`, based on a list of search candidates.
**Concept to find:** "node.name"
**Search Candidates from `mathlib`:**
--
...(candidates context)...
--
**Your Task (Follow these steps PRECISELY):**
**Step 1:  Direct Match Analysis**
- First, look for a **direct, canonical definition** among the
candidates.  A direct match is typically a `class`, `structure`, or
`def` whose name is very similar to the concept name (e.g., concept
'local ring' matches `class IsLocalRing`).
```

---

```
- If you find a clear, direct match, use that as your primary answer.
**Step 2:  Deduction from Usage Patterns (If no direct match is
found)**
- If no direct match was found in Step 1, your task is to **deduce**
the canonical name by finding a **consistent usage pattern** across
multiple `theorem` and `instance` candidates.
- **Analyze the signatures:** Look for a common identifier that is
consistently used as a **type** or **typeclass** across multiple
candidates.
- **Example:** If you are looking for "CharZero" and the search
results include `instance :  CharZero ℕ`, `instance :  CharZero ℤ`,
and `theorem my_thm [CharZero R]`, the identifier `CharZero` appears
repeatedly as a typeclass.  This is overwhelming evidence that the
canonical definition is named `CharZero`.
- **Strict Rule:** The name you select **must** be an identifier that
is explicitly present in the candidate list.  Do **not** invent,
combine, or guess a new name.  If no single, consistent pattern
emerges from the candidates, you must conclude that no confident
match can be found.
**Step 3:  Final Decision**
- Based on your analysis from Step 1 and Step 2, determine the single
best name for the concept.
- Your answer MUST be a single, valid JSON object with the following
keys:
- `"best_match"`:  The full formal name of the canonical definition
(e.g., "RingTheory.IsLocalRing").  If no confident match can be
found through either direct matching or inference, the value must
be `null`.
- `"reasoning"`:  A brief, one-sentence explanation of HOW you found
the match.  It must be one of the following strings:  "Found a direct
definition." or "Inferred from usage in instances and theorems." or
"No confident match found."
**JSON Output:**
```

### Prompt for Definition Synthesis Phase

```
You are a meticulous expert in Lean 4 and `mathlib4`.  Using the
following verified Lean 4 prerequisite definitions as context, write
the formal Lean definition for "node.name".
Your output must be a single, well-formed Lean 4 code block.  Do not
add any explanation outside the code block.
**Context from Previous Steps:**
--
...(context code)...
--
**Informal Definition of "node.name":**
...(informal description)...
**Your Task:  Write the Lean 4 `def` or `class`:**
Caution:  DO NOT use sorry to skip the value of the definition.
```

### Prompt for Statement Synthesis Phase

```
You are a meticulous expert in Lean 4 and `mathlib4`.  Your primary
goal is to translate informal mathematical statements into **correct,
idiomatic, and compilable** Lean 4 code that seamlessly integrates
with the existing Mathlib library.
Before generating the final code, you MUST follow a structured
thought process in five steps:
1.  **Deconstruct**:  Break down the informal statement into its core
mathematical components (e.g., objects, assumptions, conclusion).
```

```
2.  **Identify Mathlib Components**:  List the key Mathlib
definitions, theorems, and notations that are necessary to formalize
each component.  Guessing is not allowed; refer to known Mathlib
APIs.  For example, 'integral domain' corresponds to '[IsDomain R]',
'finitely generated module' to '[Module.Finite R M]'.
3.  **Plan the Formal Statement**:  Outline the structure of the
final theorem.  This includes defining the types (e.g., 'R M :
Type*'), typeclasses (e.g., '[CommRing R]'), variables, hypotheses,
and the goal.
4.  **Generate Final Code**:  Based on the plan, write the complete,
compilable Lean 4 code.
5.  Do not generate 'variable' declarations that are irrelevant to
the final theorem statement.  For a single theorem, prefer placing
all variables and hypotheses directly in the 'theorem''s signature
instead of using a global 'variable' block.
**Context (Newly Generated Definitions):**
--
...(newly generated definitions)...
--
**Informal Theorem to Formalize:**
...(informal statement)...
**Final Lean Theorem Statement:**
Caution:  Don't generate explicit header like 'import
Mathlib.RingTheory.Noetherian'.  Use 'import Mathlib'.  **Crucially,
you must NOT write the proof.** Your only goal is to state the
theorem correctly.  The proof block must be replaced with the 'sorry'
keyword.
```

## Prompt for Reflection

```
You are a Lean 4 expert.  The following code you previously generated
has a compilation error.
Your task is to analyze the error message and provide a corrected
version of the code.
You MUST follow this two-step process:
**Step 1:  Analysis and Correction Plan**
First, provide a brief analysis of the problem in the following
format:
1.  **Error Analysis:** [Summarize the main error message in one
sentence]
2.  **Root Cause:** [Explain the underlying reason for the error,
e.g., missing typeclass instance, type mismatch between a term and
its expected type, incorrect syntax, etc.]
3.  **Correction Plan:** [Describe the specific code change you will
make to fix the issue, e.g., "Change the typeclass constraint from
[Semiring R] to [Ring R]", "Explicitly access the underlying ideal
using .toIdeal", etc.]
**Step 2:  Corrected Lean 4 Code**
Then, provide the complete, corrected code in a single Lean code
block.  Do not change the original theorem statement, only fix the
proof or definition.
**Caution:** You are not sure about the explicit header, so DO NOT
generate explicit header like 'import Mathlib.RingTheory.Noetherian',
USE 'import Mathlib'.
**Crucially, you must NOT write the proof.** Your only goal is to
state the theorem correctly.
**Failed Code:**
...(previous code)...
**Error Message from Lean Compiler:**
...(error message)...
```

```
Provide the complete, corrected Lean 4 code in a single code block,
without any extra explanation.  USE 'import Mathlib' as a header!
```

# E  GENERALIZATION ACROSS MATHEMATICAL DOMAINS

While our primary experimental analysis centers on the FATE algebra datasets, the mechanisms underlying Aria are not intrinsically limited to algebra. In this section, we analyze the system's performance across diverse mathematical fields and discuss the rationale behind our domain selection.

## E.1  PERFORMANCE ON PROOFNET

To empirically validate domain generalization, we break down the performance on the ProofNet benchmark by subfield. As shown in Table 6, Aria demonstrates high consistency across undergraduate-level algebra, analysis, number theory and topology. Furthermore, it surpasses the strong baseline (Goedel-V2) in every category. Notably, in number theory and topology, Aria achieves a significant margin in final accuracy, suggesting that its retrieval and planning capabilities are robust to domain shifts.

Table 6: Performance breakdown by domain on the ProofNet benchmark. Aria demonstrates consistent superiority over the Goedel-V2 baseline across all subfields.

| Metric | Algebra | Analysis | Number Theory | Topology |
|---|---|---|---|---|
| Aria (Ours) Compiler | **97.4%** | **100.0%** | **100.0%** | **96.7%** |
| Aria (Ours) Final Acc. | **64.7%** | **64.8%** | **71.4%** | **56.7%** |
| Goedel Compiler | 54.7% | 81.8% | 90.5% | 26.7% |
| Goedel Final Acc. | 28.9% | 44.3% | 47.6% | 11.7% |

## E.2  CASE STUDY: BOREL'S CONJECTURE IN TOPOLOGY

Furthermore, we successfully applied Aria to formalize Borel's Conjecture in topology. This task requires handling distinct mathematical structures (e.g., `Manifold`, `ChartedSpace`, `HomotopyGroup`) that differ significantly from algebraic rings and modules.

This successful formalization confirms that Aria's GoT planner can effectively navigate the definition dependencies in non-algebraic domains, provided that the underlying library support exists.

```
import Mathlib

/-- A closed manifold is a compact manifold with empty boundary. -/
class IsClosedManifold {𝕜 : Type*} [NontriviallyNormedField 𝕜] {E :
    Type*}
  [NormedAddCommGroup E] [NormedSpace 𝕜 E] {H : Type*} [TopologicalSpace
    H]
  (I : ModelWithCorners 𝕜 E H) (n : WithTop ℕ∞) (M : Type*)
    [TopologicalSpace M]
  [ChartedSpace H M] extends IsManifold I n M, CompactSpace M : Prop
    where
  /-- The boundary of a closed manifold is empty. -/
  boundaryless : {x : M | I.IsBoundaryPoint (chartAt H x x)} = ∅

/-- An aspherical topological manifold is a topological manifold `M`
    that is path-connected and for which the `k`-th homotopy group
    `πₖ(M,x)` is trivial for all `k ≥ 2` and all basepoints `x : M`. -/
structure IsAsphericalTopologicalManifold
    {𝕜 : Type*} [NontriviallyNormedField 𝕜]
    {E : Type*} [NormedAddCommGroup E] [NormedSpace 𝕜 E]
```

```
     {H : Type*} [TopologicalSpace H] (I : ModelWithCorners 𝕜 E H)
     (n : WithTop ℕ∞) (M : Type*) [TopologicalSpace M] [ChartedSpace H
  M] : Prop where
  /-- An aspherical topological manifold is a topological manifold. -/
  is_manifold : IsManifold I n M
  /-- An aspherical topological manifold is path-connected. -/
  path_connected : PathConnectedSpace M
  /-- The `k`-th homotopy group of an aspherical topological manifold is
     trivial for `k ≥ 2`. -/
  homotopy_groups_trivial (k : ℕ) (hk : 2 ≤ k) (x : M) :
     Subsingleton (HomotopyGroup (Fin k) M x)

/-- Let M and N be closed and aspherical topological manifolds. If
    f : M → N is a homotopy equivalence, then f is homotopic to a
    homeomorphism. -/
theorem borel_conjecture_for_topological_manifolds
    {𝕜 : Type*} [NontriviallyNormedField 𝕜]
    {E : Type*} [NormedAddCommGroup E] [NormedSpace 𝕜 E]
    {H : Type*} [TopologicalSpace H]
    {I : ModelWithCorners 𝕜 E H}
    {n : WithTop ℕ∞}
    {M : Type*} [TopologicalSpace M] [ChartedSpace H M]
    [IsClosedManifold I n M]
    (hM_aspherical : IsAsphericalTopologicalManifold I n M)
    {N : Type*} [TopologicalSpace N] [ChartedSpace H N]
    [IsClosedManifold I n N]
    (hN_aspherical : IsAsphericalTopologicalManifold I n N)
    (f : ContinuousMap.HomotopyEquiv M N) :
    ∃ (g : M ≃ₜ N), ContinuousMap.Homotopic f.toFun (g : C(M, N)) := by
  sorry
```

# F    STATEMENT ON THE USE OF LARGE LANGUAGE MODELS (LLMS)

In accordance with the policy, we disclose that Large Language Models (LLMs) played a significant role in the preparation of this manuscript. The authors take full responsibility for all content, including any text generated by these models, and have meticulously reviewed and edited all outputs for accuracy, originality, and scientific integrity.

We utilized Google's Gemini-2.5-Pro as a language editing tool. Its role was strictly limited to improving clarity, correcting grammatical errors, and rephrasing sentences.

