# OpenReview forum: "Aria: an Agent for Retrieval and Iterative Auto-Formalization via Dependency Graph"
_ICLR.cc/2026/Conference — ICLR 2026 Poster_

### Official Review · Reviewer_dKZk · 2025-10-28

**Soundness:** 2
**Presentation:** 3
**Contribution:** 3
**Rating:** 6
**Confidence:** 4

**Summary:**

This paper proposes Aria, a comprehensive agent-based autoformalization pipeline designed to emulate human reasoning for research-level mathematics. The system first performs graph-of-thought decomposition, transforming natural language statements into dependency graphs and grounding the identified concepts in mathlib when possible. For ungrounded concepts, it employs graph-of-thought synthesis to generate Lean representations through a self-reflective loop.

The authors further introduce AriaScorer, which decomposes informal statements into atomic assumptions and conclusions, then aligns them with formal clauses retrieved from Lean’s library and their corresponding natural language versions to compute an aggregated correctness score.

Experimental results demonstrate that Aria significantly outperforms existing LLM-based autoformalization baselines on the ProofNet, FATE benchmarks, and real-world conjectures. Moreover, AriaScorer achieves more accurate verification of formalization correctness compared to other evaluation methods.

**Strengths:**

* The paper is well-written and well-motivated. Its focus on research-level mathematical autoformalization is interesting.
* The proposed Aria and AriaScorer frameworks are novel, integrating several intuitively and carefully designed components into a cohesive pipeline.
* Experimental results demonstrate that both Aria and AriaScorer are effective, outperforming existing methods by a substantial and consistent margin.
* The key insights underlying AriaScorer are also inspiring, offering a new perspective on evaluating the correctness of autoformalization.

**Weaknesses:**

There is no human evaluation in the main results, which makes the reported performance less reliable. Moreover, since AriaScorer uses Gemini for both generation and evaluation, the results may be biased toward its own generated responses, especially when compared to other baselines such as Goedel-Autoformalizer.

**Questions:**

* Could you include some manual evaluation on the main experiments to better assess the quality and reliability of the autoformalized results?

---

> ### Author Response · Authors · 2025-11-24
> **Response to Reviewer dKZk**
>
> We sincerely thank the reviewer for this critical observation regarding evaluation reliability. We fully acknowledge the potential risk of bias when using the same backbone model for generation and evaluation. We address this with the following human verification measures:
>
> 1. **Addressing Bias via Human Verification**
>
> To rigorously quantify bias, we constructed an expert-verified ground truth dataset for FATE-X. Crucially, the metrics in Table 2 allow us to calculate the true accuracy of the autoformalized statements based directly on human labels. At the threshold 0.9, AriaScorer achieves a precision of 95.5%, confirming its high reliability as a filter. While human experts identified 52 correct solutions (TP + FN), AriaScorer accepted 44 (TP + FP). Consequently, our reported performance (44.0%) serves as a rigorous lower bound for true performance (52.0%), rather than an inflated estimate. This effectively demonstrates that the scorer is conservative and not biased toward favoring its own generator.
>
> 2. **Manual Verification of Main Results**
>
> In addition to the FATE-X analysis, we explicitly performed manual verification for the Conjectures dataset. Under strict human evaluation, Aria achieves 42.9%, while Goedel-V2 achieves 0%. This confirms that the performance gap is a result of genuine architectural superiority, specifically the synthesis capability, rather than metric bias.
> We will include these verification details prominently in the final version to substantiate the reliability of our claims.

---

### Official Review · Reviewer_tTzy · 2025-10-31

**Soundness:** 3
**Presentation:** 3
**Contribution:** 3
**Rating:** 6
**Confidence:** 3

**Summary:**

The paper introduces Aria, a system designed to perform autoformalization of high-level mathematical statements into Lean 4 syntax. Aria emulates human reasoning via a two-phase Graph-of-Thought (GoT) process: (1) recursively decomposing a conjecture into a dependency graph of definitions and sub-statements, and (2) constructing Lean-formalized statements using retrieved definitions from Mathlib. To enforce semantic precision, it employs AriaScorer, a retrieval-based verifier that grounds terms and checks logical consistency.
The model combines retrieval-augmented generation, graph reasoning, and self-reflection for improved correctness. On three benchmarks (ProofNet, FATE-X, and Homological Conjectures) Aria achieves strong improvements. The paper claims Aria is the first system to synthesize novel definitions during autoformalization, addressing a key bottleneck in automated mathematical reasoning.

**Strengths:**

1. The combination of retrieval, graph-of-thought reasoning, and verification is well-motivated and logically coherent. Autoformalization of research-level statements is a frontier problem, and this paper is impactful both for Lean and general symbolic reasoning research.

2. Performance is satisfactory. The system achieves SOTA results across diverse benchmarks, especially on difficult tasks where all prior models can't do well.

3. It tackles hallucination and definition synthesis, the two well-known weaknesses of current formalization LLMs.

4. The use of term-level retrieval (AriaScorer) for semantic grounding is a technically meaningful contribution that enhances reliability.

**Weaknesses:**

1. More ablation study can make it even better. The contribution of each submodule (retrieval, GoT reasoning, reflection, AriaScorer) is not quantified in detail; ablation or controlled comparison would strengthen claims.

2. Some discussion on Aria's generality to other systems (e.g., Coq, Isabelle, Lean 3) will make it more impactful.

3. While promising, the "first to synthesize novel definitions" claim may need stronger empirical evidence (e.g., human evaluation verifying novelty and correctness).

4. More detailed discussion on computational cost or efficiency will be very beneficial for assessing scalability.

**Questions:**

Addressing the points I mentioned in weakness part should be enough.

---

> ### Author Response · Authors · 2025-11-24
> **Response to Reviewer tTzy (1/3)**
>
> **R W1: Ablation Study on Submodules**
>
> We sincerely thank the reviewer for this constructive suggestion. We agree that a detailed quantification of each submodule is crucial for validating our architecture. As shown in the ablation studies in Section 4.4 and Appendix C, we have isolated these contributions. In practice, removing the reflection component causes performance to drop from 44.0% to 14.0% on FATE-X. Similarly, ablating the GoT planner reduces success on Conjectures from 42.9% to 7.1%, while removing RAG causes the success rate to fall to 0%. Furthermore, as shown in Table 2, our term-level grounding in AriaScorer improves accuracy from 73.9% to 82.6% on FATE-X. These comparisons effectively validate the necessity of our architectural choices.
>
> **R W2: Generalization to Other Systems**
>
> We sincerely thank the reviewer for this thoughtful suggestion. We agree that discussing Aria's generality is valuable. Generally, Aria is composed of several universal components, including the retriever, GoT planner, and compiler-based reflector. Each of these is swappable and adaptable to mature formal languages such as Coq or Isabelle. The GoT planner utilizes the LLM's mathematical reasoning to decompose concepts, a process operating on conceptual definitions independent of specific syntax. Furthermore, the reflection loop relies on standard compiler feedback, a mechanism fundamental to all interactive theorem provers. Consequently, adapting Aria to other systems primarily involves switching the retrieval corpus and compiler interface. We look forward to applying our framework to a broader range of formal languages in future work.
>
> **R W3: Verification of Novel Definitions**
>
> We sincerely thank the reviewer for this rigorous scrutiny. We agree that the claim regarding the synthesis of "novel definitions" requires stronger empirical evidence. We address this through manual inspection and performance analysis.
>
> First, such evidence is established through careful manual inspection. We carefully examined a large number of generated cases. We confirmed that specific definitions synthesized by Aria were indeed absent from the Mathlib commit used in our experiments (Mathlib v4.16.0). Below, we provide 2 of these representative cases with synthesized novel definitions generated by our system as well as our justifications for their novelty.
>
> 1. Definition of Aspherical Topological Manifold
>
> **Formalized statement**:
> ```lean
> import Mathlib
>
> /-- A closed manifold is a compact manifold with empty boundary. -/
> class IsClosedManifold {k : Type*} [NontriviallyNormedField k] {E : Type*}
>   [NormedAddCommGroup E] [NormedSpace k E] {H : Type*} [TopologicalSpace H]
>   (I : ModelWithCorners k E H) (n : WithTop ℕ∞) (M : Type*) [TopologicalSpace M]
>   [ChartedSpace H M] extends IsManifold I n M, CompactSpace M : Prop where
>   /-- The boundary of a closed manifold is empty. -/
>   boundaryless : {x : M | I.IsBoundaryPoint (chartAt H x x)} = ∅
>
> /-- An aspherical topological manifold is a topological manifold `M` that is path-connected and for which the `k`-th homotopy group `πₖ (M, x)` is trivial for all `k ≥ 2` and all basepoints `x : M`. -/
> structure IsAsphericalTopologicalManifold
>     {k : Type*} [NontriviallyNormedField k]
>     {E : Type*} [NormedAddCommGroup E] [NormedSpace k E]
>     {H : Type*} [TopologicalSpace H] (I : ModelWithCorners k E H)
>     (n : WithTop ℕ∞) (M : Type*) [TopologicalSpace M] [ChartedSpace H M] : Prop where
>   /-- An aspherical topological manifold is a topological manifold. -/
>   is_manifold : IsManifold I n M
>   /-- An aspherical topological manifold is path-connected. -/
>   path_connected : PathConnectedSpace M
>   /-- The `k`-th homotopy group of an aspherical topological manifold is trivial for `k ≥ 2`. -/
>   homotopy_groups_trivial (k : ℕ) (hk : 2 ≤ k) (x : M) :
>     Subsingleton (HomotopyGroup (Fin k) M x)
>
> /-- Let $M$ and $N$ be closed and aspherical topological manifolds. If $f \colon M \to N$ is a homotopy equivalence, then $f$ is homotopic to a homeomorphism. -/
> theorem borel_conjecture_for_topological_manifolds
>     {k : Type*} [NontriviallyNormedField k]
>     {E : Type*} [NormedAddCommGroup E] [NormedSpace k E]
>     {H : Type*} [TopologicalSpace H]
>     {I : ModelWithCorners k E H}
>     {n : WithTop ℕ∞}
>     {M : Type*} [TopologicalSpace M] [ChartedSpace H M]
>     [IsClosedManifold I n M]
>     (hM_aspherical : IsAsphericalTopologicalManifold I n M)
>     {N : Type*} [TopologicalSpace N] [ChartedSpace H N]
>     [IsClosedManifold I n N]
>     (hN_aspherical : IsAsphericalTopologicalManifold I n N)
>     (f : ContinuousMap.HomotopyEquiv M N) :
>     ∃ (g : M ≃ₜ N), ContinuousMap.Homotopic f.toFun (g : C(M, N)) := by
>   sorry
> ```

---

> > ### Author Response · Authors · 2025-11-24
> > **Response to Reviewer tTzy (2/3)**
> >
> > (cont.)
> >
> > **Justification**:
> > For instance, regarding the synthesized structure `IsAsphericalTopologicalManifold`, we confirmed that Mathlib lacks a unified definition for this concept. Aria detected this gap and autonomously synthesized a new structure by aggregating `IsManifold`, `PathConnectedSpace`, and the precise homotopy constraint `Subsingleton (HomotopyGroup (Fin k) M x)` for $k \ge 2$. This synthesis accurately translates the informal topological requirements into rigorous formal code, demonstrating both the system's capacity to generate novel definitions absent from the library and its adherence to semantic correctness.
> >
> > 2. Definition of Regular Ring
> >
> > **Formalized statement**:
> > ```lean
> > /-- A commutative ring `R` is a regular local ring if it is a Noetherian local ring, and the Krull dimension of `R` is equal to the dimension of its cotangent space `m/m²` as a vector space over the residue field `R/m`. -/
> > class IsRegularLocalRing (R : Type*) [CommRing R] [IsLocalRing R] : Prop where
> >   isNoetherian : IsNoetherianRing R
> >   dim_eq_finrank_cotangent :
> >     (letI := isNoetherian; ringKrullDim R) =
> >     ↑(FiniteDimensional.finrank (IsLocalRing.ResidueField R) (IsLocalRing.CotangentSpace R))
> >
> > /-- A commutative ring `R` is a regular ring if it is a Noetherian ring, and for every prime ideal `p`, the localization `Rₚ` at `p` is a regular local ring. -/
> > class IsRegularRing (R : Type*) [CommRing R] [IsNoetherianRing R] : Prop where
> >   isRegularLocal_at_prime :
> >     ∀ (p : PrimeSpectrum R), IsRegularLocalRing (Localization.AtPrime p.asIdeal)
> > ```
> >
> > **Justification**: Similarly, regarding the `IsRegularRing` case, we confirmed that Mathlib lacks these specific bundled classes. Aria demonstrated advanced hierarchical planning by synthesizing this concept through a dependency chain. First, it defines `IsRegularLocalRing` by correctly equating the Krull dimension to the vector space dimension of the cotangent space m/m². Crucially, it then utilized this newly synthesized component to define the global `IsRegularRing` structure, accurately formalizing the condition that localizations at all prime ideals must satisfy the local regularity predicate. This validates the system's ability to generate mathematically rigorous, multi-step dependency chains absent from the library.
> >
> > Second, to make all those manual observations to the ground, the new definition synthesis ability of our model is further proved by end-to-end benchmark result. In our paper, conjectures dataset is specifically chosen for its high difficulty, requiring numerous novel definitions. Aria's ability to handle these missing concepts is explicitly demonstrated in the end-to-end benchmark results. While baselines without these capabilities scored 0%, Aria achieved 42.9% accuracy. This performance gap strongly supports our system's capability to generate valid definitions where the bare mathlib library is insufficient.
> >
> > **R W4: Computational Cost Analysis**
> >
> > We agree with the reviewer that cost analysis is crucial for assessing the scalability and practical value of systems employing Large Language Models (LLM) in an agent workflow. Our design goal was to achieve outstanding performance while maintaining a competitive cost profile.
> >
> > As noted in our comparison of inference efficiency with Goedel-V2 (L355), the average number of Gemini-2.5-Pro API calls required for theorem and definition generation on the FATE-X dataset is 17.7 times. However, the total system overhead encompasses more than just final generation; it includes decomposition, search, and candidate selection at every node. We break down the system costs below:
> >
> > | Component | Model Selection | Frequency and Overhead Analysis |
> > | :--- | :--- | :--- |
> > | **Retrieval in Mathlib** | LeanSearch. | This is a simple tool invocation; its cost is orders of magnitude lower than any LLM API call. |
> > | **Candidate Selection** | High Reasoning Capability (e.g. Gemini-2.5-Pro, GPT-5, DeepSeek-R1). | Each internal node necessitates one check. Since the token usage for each query is low, the cumulative cost is negligible even for the most complex conjectures. |
> > | **Node Expansion** | High Reasoning Capability (e.g. Gemini-2.5-Pro, GPT-5, DeepSeek-R1). | Requires only one API call per internal node and the root node. Given that the token consumption per call is minimal, the overall cost per problem remains low. |
> > | **Theorem and Definition Generation** | High Reasoning Capability (e.g. Gemini-2.5-Pro, GPT-5, DeepSeek-R1). | This is the main cost sink, due to the token consumption involved in each reflection step. |

---

> > > ### Author Response · Authors · 2025-11-24
> > > **Response to Reviewer tTzy (3/3)**
> > >
> > > (cont.)
> > >
> > > Since Theorem and Definition Generation stage is the primary cost driver, we analyzed its scaling behavior. Higher difficulty implies deeper hierarchical relationships, requiring more GoT nodes and reflection cycles. Consequently, formalizing a statement in the Conjectures dataset requires over 10x the API calls of the High School Contest dataset, as shown in the table below.
> > >
> > > | Benchmark Dataset | Difficulty Level | Average API Calls (Gemini-2.5-Pro) |
> > > | :--- | :--- | :--- |
> > > | High School Contest (e.g. CombiBench) | Simple | ~2-3 calls |
> > > | Undergraduate Exercises (e.g. ProofNet) | Medium | ~4-5 calls |
> > > | FATE-H | Mid-high | ~5-7 calls |
> > > | FATE-X | High | ~17-19 calls |
> > > | Conjectures | Extra High | > 30 calls|
> > >
> > > Formalizing high-difficulty, deeply structured mathematical statements is a highly specialized task. Producing formal statements at the scale and complexity of FATE-X and Conjectures via human experts entails prohibitive labor costs. Our analysis indicates that a qualified expert requires 1–2 hours per problem (about \\$50 labor cost), whereas our pipeline achieves formalization in roughly 10 minutes per problem at a fraction of the cost (\\$3–5 API cost). Furthermore, our system supports massive parallelization. By comparison, our system largely reduces this cost barrier, offering a scalable solution for research-level formalization.
> > >
> > > To summarize, our cost analysis proves that Aria is a cost-aware framework. By delegating low-cost tasks to simpler models and concentrating the expensive LLM overhead on the GoT framework’s critical reasoning and generation steps, we effectively avoid the "brute-force search" and inefficient trial-and-error common in generic LLM agent systems. This substantiates our claim that ARIA is a computationally efficient solution for research-level autoformalization.

---

### Official Review · Reviewer_Tkra · 2025-10-31

**Soundness:** 2
**Presentation:** 2
**Contribution:** 2
**Rating:** 4
**Confidence:** 3

**Summary:**

This paper presents Aria, an agent system for auto-formalizing mathematical statements through a two-phase Graph-of-Thought (GoT) process: recursively decomposing statements into conceptual dependency graphs anchored to Mathlib via RAG, then bottom-up construction of formalizations synthesizing new definitions through compiler-in-the-loop reflection. The paper also introduces AriaScorer, a semantic checker achieving term-level grounding by retrieving actual definitions of Lean terms. Experiments demonstrate that Aria achieves 44.0% accuracy on FATE-X (baseline: 24.0%) and 42.9% on Homological Conjectures (all baselines: 0%), significantly outperforming existing methods.

**Strengths:**

- **Effective system architecture**: The paper successfully integrates GoT planning, RAG, and compiler-in-the-loop reflection to address the critical failure point of prior models in synthesizing new definitions. Ablation studies in Table 3 validate the necessity of each component: removing RAG or reflection causes accuracy to collapse from 42.9% to 0% on the Conjectures dataset.

- **Innovative contribution of AriaScorer**: This checker transcends text similarity limitations by employing the static analyzer jixia to retrieve actual definitions of Lean terms. Table 2 validates that this term-level grounding step is crucial for improving accuracy.

- **Breakthrough experimental results**: The system achieves 42.9% success rate on the Homological Conjectures benchmark where all baseline models completely fail (scoring 0%), demonstrating the unique capability and practical value of this compound approach for conjecture-level formalization.

**Weaknesses:**

- **Compound propagation risk of semantic errors**: The paper explicitly states that the GoT synthesis phase ensures only syntactic correctness and "cannot preclude correctly-typed but semantically wrong translations," while AriaScorer's semantic checking is described as a final step (Figure 1(C)). This creates a critical logical gap: if a base concept (e.g., "Cohen-Macaulay Module" in Figure 1) is semantically incorrect during synthesis, it will serve as a syntactically valid but semantically flawed premise for synthesizing all dependent parent nodes, invalidating entire dependency branches. The paper lacks crucial discussion or experimental validation on whether AriaScorer should be iteratively applied to each node during the synthesis phase to prevent this issue.

- **Insufficient explanation of contradictory phenomena in GoT ablation**: Table 5 shows that removing GoT paradoxically increases compilation rate (89%→95%) while decreasing final accuracy (71%→54%) on FATE-H. Although Section C.2 mentions this leads to "synthesis failure" and "interface hallucination" failure modes, it inadequately explains why a simpler monolithic approach systematically produces more syntactically simple (thus easier to compile) but semantically incorrect code. The fundamental mechanism of this "high compilation, low accuracy" failure mode remains unclear, weakening understanding of GoT's core value.

**Questions:**

1. **Regarding semantic error propagation mechanism**: Is AriaScorer's semantic checking iteratively applied to each newly synthesized node during the GoT synthesis phase (Figure 1(B)), or only performed once at the end? If not iteratively applied, how does the system prevent or recover from compound semantic errors caused by intermediate definitions that are "correctly-typed but semantically wrong"? Is there experimental data quantifying the impact and system robustness under such error scenarios?

2. **Regarding the deep mechanism of Table 5's anomaly**: Removing GoT increases compilation rate (89.0%→95.0%) but decreases accuracy (71.0%→54.0%) on FATE-H. Does this suggest that the non-GoT monolithic approach tends to produce syntactically simpler (thus easier to compile) but semantically incorrect formalizations? Can you provide concrete cases for in-depth analysis of the root cause of this "high compilation, low accuracy" failure mode? What does this reveal about GoT's core value (enforcing semantic structure vs. syntactic simplification)?

3. **Regarding reliability of critical decision points in GoT decomposition**: This process relies on an LLM reasoner to judge whether retrieved Mathlib candidates are "suitable matches" and decide whether to use candidates or trigger synthesis of new concepts. Overall pipeline performance critically depends on this decision's accuracy. How is this LLM reasoner's accuracy evaluated? Is there data showing the specific impact of its erroneous decisions (e.g., deciding to synthesize an existing concept or incorrectly matching an irrelevant concept) on end-to-end performance?

4. **Regarding robustness boundaries of AriaScorer**: This checker's term-level grounding depends on retrieving term information from the Herald dataset. What are AriaScorer's failure modes if Lean terms used in formalized statements are not included in this dataset, or if informal descriptions of terms in the dataset are inaccurate or ambiguous? Does the paper experimentally quantify the checker's robustness to incomplete grounding datasets? How does this affect the method's generalization capability to new domains or rapidly evolving Mathlib versions?

---

> ### Author Response · Authors · 2025-11-24
> **Response to Reviewer Tkra (1/3)**
>
> **R W1/Q1: Clarification on Verification Strategy**
>
> We thank the reviewer for highlighting the risk of semantic error propagation. To clarify, AriaScorer is applied only at the final stage. We deliberately designed the system this way based on specific engineering observations and empirical data.
>
> This architectural decision was driven by a strategic trade-off between verification rigor, algorithmic stability, and computational efficiency. Below, we explain the reasons behind this design and provide empirical evidence of its robustness.
>
> 1. Fundamentally, AriaScorer is engineered to function as a rigorous, independent checker rather than a lightweight feedback signal. This process involves multiple LLM calls and database searches to check individual definitions. Because this process is rigorous and computationally expensive, AriaScorer is best suited as a final validation step to ensure the quality of the completed solution, rather than being run repeatedly during generation.
>
> 2. In our early experiments, we tried applying semantic feedback iteratively, but we observed two main issues:
>
>   - Instability: We found that correcting a semantic issue frequently disrupted the syntactic structure, leading to oscillatory behavior.
>
>   - Low inference efficiency: Since AriaScorer is computational intensive, applying it to every intermediate node would drastically increase the total computation time. This inefficiency is further compound by the oscillatory behavior.
>
> 3. We address the reviewer's concern regarding semantic error propagation from two complementary perspectives: the low frequency of such propagation and the rigor of the final verification. Our experimental data suggests that errors caused by "correctly typed but semantically wrong" intermediate definitions are statistically rare. We conducted a manual root cause analysis of all failure cases in the FATE-X dataset. The analysis revealed that only a single instance of failure was attributable to a flaw in definition synthesis that propagated to cause the final semantic inconsistency. Here is full code for the case, along with the precise error analysis:
>
> ```lean
> import Mathlib
>
> universe u
>
> /--
> A ring `R` is a catenary ring if it is a commutative Noetherian ring, and for any two
> prime ideals `p` and `q` with `p ⊆ q`, all saturated chains of prime ideals between
> `p` and `q` have the same length.
>
> The length of a finite chain of prime ideals `p₀ ⊂ p₁ ⊂ ... ⊂ pₙ` is `n`. This corresponds
> to the chain having `n+1` elements. Thus, two chains have the same length if and only
> if they have the same number of elements (cardinality).
> -/
> class CatenaryRing (R : Type u) [CommRing R] extends IsNoetherianRing R : Prop where
>   /--
>   For any two prime ideals `p, q` with `p ≤ q`, all saturated (i.e. maximal) chains
>   of prime ideals in the interval `Set.Icc p q` have the same cardinality.
>   -/
>   eq_saturated_chain_length :
>     ∀ (p q : PrimeSpectrum R), p ≤ q →
>       ∀ (c₁ c₂ : Set (Set.Icc p q)),
>         IsMaxChain (· ≤ ·) c₁ →
>         IsMaxChain (· ≤ ·) c₂ →
>         Set.ncard c₁ = Set.ncard c₂
>
> /--
> A Noetherian Unique Factorization Domain of Krull dimension at most 3 is catenary.
> -/
> theorem ufd_of_dim_le_three_is_catenary (A : Type u)
>     [CommRing A] [IsDomain A] [IsNoetherianRing A] [UniqueFactorizationMonoid A]
>     (h_dim : Order.krullDim (PrimeSpectrum A) ≤ 3) :
>     CatenaryRing A :=
>   sorry
> ```
>
> Below is the error analysis provided by AriaScorer:
> ```markdown
> **A is a catenary ring**:
> - Math: $A$ is a catenary ring, i.e., for any two prime ideals $P, Q$ of $A$ with $P \subseteq Q$, any two saturated chains of prime ideals between $P$ and $Q$ have the same length.
> - Lean: `CatenaryRing A`. The provided `class CatenaryRing` definition `extends IsNoetherianRing R`, which means that to be a `CatenaryRing`, a ring must be Noetherian in addition to satisfying the chain condition. The mathematical definition provided does not require the ring to be Noetherian. This makes the Lean conclusion stronger than the mathematical one.
> - Match: Major inconsistency.
> ```
>
> This specific case involves the formalization of the "Catenary Ring" property. The GoT planner synthesized a class `CatenaryRing` that explicitly extends `IsNoetherianRing`. However, the definition of a catenary ring (i.e. saturated chains have equal length) is independent of the Noetherian property. Even though the context of this specific problem explicitly includes the Noetherian ring condition, AriaScorer still judged the synthesized definition which hallucinated the `extends IsNoetherianRing` clause as erroneous during the final check. This demonstrates that AriaScorer is sufficiently rigorous in its final inspection. Furthermore, if we were to utilize AriaScorer as a feedback signal for reflection in addition to its role as the final arbiter, the system would learn the scorer's preferences during iteration. This would induce self-referential bias.

---

> > ### Author Response · Authors · 2025-11-24
> > **Response to Reviewer Tkra (2/3)**
> >
> > (cont.)
> >
> > In summary, we recognize that incorporating an efficient, real-time semantic signal within the iterative loop holds potential for enhancing system performance. However, this distinct utility differs from the primary design objective of AriaScorer, which is engineered as a high-accuracy terminal evaluator.
> >
> > Nonetheless, we acknowledge that the risk of error propagation correlates with the depth of dependency graph. While the GoT dependency graph in FATE-X is relatively shallow (averaging 2-3 layers), this issue may become more pronounced in future, more complex conjectures involving large-scale dependency graphs.  Looking ahead to large-scale formalization challenges (e.g., full textbooks), we value the reviewer’s suggestion. Our future work may aim to efficiently integrate semantic signals into larger systems by adopting strategoc checkpointing (e.g., every $k$ layers).
> >
> > **R W2/Q2: Analyzing the "High Compilation, Low Accuracy" Anomaly**
> >
> > To provide concrete evidence of the "high compilation, low accuracy" anomaly, we include an illustrative case from the FATE-H benchmark. The case demonstrates how the synthesis of a semantically precise definition (with GoT) carries a higher syntactic risk.
> >
> > Below, we present the code generated by the full Aria system and the 'Aria w/o GoT' ablation.
> >
> > This is the code generated by the Aria full system:
> > ```lean
> > import Mathlib
> > open Polynomial
> > private noncomputable abbrev F_seven := ZMod 7
> > private noncomputable abbrev p_over_F_seven : Polynomial F_seven := X ^ 3 - C 2
> > private noncomputable abbrev K_ext := AdjoinRoot p_over_F_seven
> > theorem prime_ideals_of_tensor_product_are_principal :
> >   ∀ (I : Ideal (K_ext ⊗[F_seven] K_ext)), I.IsPrime → I.IsPrincipal :=
> >     sorry
> > ```
> >
> > This is the code generated by the 'Aria w/o GoT' ablation.
> > ```lean
> > import Mathlib
> > open Polynomial TensorProduct
> > /--The prime ideals of the tensor product algebra `𝔽₇[α] ⊗[𝔽₇] 𝔽₇[α]`,
> > where `α` is a root of `X³ - 2`, are all principal.
> > -/
> > theorem prime_ideals_of_F7_adjoin_tensor_self_are_principal :
> >   let F := ZMod 7
> >     let p : Polynomial F := X ^ 3 - C 2
> >       let K := AdjoinRoot p
> >         let R := K ⊗[F] K
> >           ∀ (P : Ideal R), P.IsPrime → P.IsPrincipal := by
> >             sorry
> > ```
> >
> > In this case, the GoT planner prioritizes semantic explicitness and structural completeness by forcing the generation of all necessary intermediate definitions (`K_ext`, `p_over_F_seven`, etc.). This structural complexity significantly increases the total attack surface for simple errors like global namespace management (which cause failure of the GoT version). Moreover, Lean's type class mechanism complicates this modularity: instances of equivalent but distinct definitions are often not interoperable. Consequently, reusing upstream definitions frequently fails verification, forcing the system to construct explicit equivalence proofs rather than relying on direct matching. The performance penalty is particularly noticeable on simpler benchmarks like FATE-H, as the lack of necessity for complex hierarchical planning means the increased setup overhead (i.e., longer generated code) only contributes to a reduced compile rate.
> >
> > However, this relationship shifts on structurally complex benchmarks like FATE-X. Our data shows that the compilation success rate is identical (69%) both with and without the GoT planner. This suggests that on harder datasets, the structural benefit derived from GoT's decomposition utility effectively offsets the syntactic cost caused by longer generated codes. Moreover, the final accuracy still drops significantly without GoT, which again shows the critical role of GoT in ensuring semantic consistency.
> >
> > Hence the 'high compilation, low accuracy' phenomenon observed solely on the FATE-H dataset does not capture the core value of GoT. This is because the vast majority of FATE-H problems require no prerequisite definitions for formalization, and GoT is specifically designed for high-difficulty, research-level statements. By simulating the cognitive process of human mathematicians building complex theories, the Graph-of-Thought (GoT) framework is intended to fully leverage the powerful natural-language mathematical knowledge of reasoning LLMs to expand their formal mathematical capability.
> >
> > All prior auto-formalization works, with or without RAG, rely on the model's static formal knowledge coupled with the existing formal math library (Mathlib). Consequently, they always fail when formalizing statements that require prerequisite definitions absent from Mathlib. This represents a significant underutilization of the natural-language mathematical capability of current large reasoning models. Through a novel algorithm, GoT explicitly leverages this capability by constructing a dynamic dependency graph. This effectively realizes a breakthrough in the model's formal-language mathematical capabilities.

---

> > > ### Author Response · Authors · 2025-11-24
> > > **Response to Reviewer Tkra (3/3)**
> > >
> > > (cont.)
> > >
> > > As difficulty increases—from FATE-H to FATE-X to Conjectures—the 'high compilation' anomaly gradually diminishes. Throughout this spectrum, the GoT-enhanced system consistently maintains a lead in final accuracy, culminating in outstanding performance on the Homological Conjectures dataset, which definitively validates the value of our system.
> > >
> > > **R W3: Reliability Assessment of the Decision Reasoner**
> > >
> > > We thank the reviewer for pinpointing this critical decision mechanism. We address the evaluation and specific impact of this component by distinguishing between two error categories.
> > >
> > > 1. Retrieval Omission: The correct term was not retrieved by LeanSearch. This represents an external limitation of the retrieval tool and falls outside the scope of this paper.
> > > 2. Reasoning Error: The correct term was retrieved but misidentified by the LLM. To evaluate this, we conducted a detailed manual inspection of all failure cases in the FATE-X dataset. Crucially, among the 45 actual error cases, we found zero instances where the formalization failure was caused by the LLM reasoner making an incorrect judgment (i.e., erroneously matching an irrelevant concept or failing to identify an essential one). This empirical evidence demonstrates that the LLM reasoner step is robust and not the bottleneck for overall pipeline performance.
> > >
> > > **R W4: Analysis of AriaScorer's Grounding Robustness**
> > >
> > > We sincerely thank the reviewer for examining AriaScorer's robustness. First, regarding data completeness, the Herald dataset is a comprehensive corpus for Mathlib, ensuring coverage of core definitions. Rare missing terms are typically minor concepts that do not hinder LLM's reasoning. Second, regarding ambiguity, our end-to-end manual verification confirms that vague informal descriptions do not cause failures. This resilience stems from AriaScorer prioritizing formal verification. By retrieving authoritative formal definitions from Mathlib as the ground truth, the system overrides potential ambiguity in the informal description. Crucially, as shown in Table 2, AriaScorer achieves a high precision of 95.5% and largely outperforms LeanScorer. This confirms that potential data gaps do not negatively impact the system's stability, ensuring the reliability required for generalization.

---

> > > > ### Comment · Reviewer_Tkra · 2025-11-25
> > > >
> > > > The authors' rebuttal has addressed most of my concerns. I will raise my score accordingly. I hope the authors will update the manuscript as discussed.

---

### Official Review · Reviewer_ZfvQ · 2025-10-31

**Soundness:** 2
**Presentation:** 2
**Contribution:** 2
**Rating:** 4
**Confidence:** 3

**Summary:**

The paper presents ARIA, an agent for auto-formalization of mathematical statements in Lean 4. ARIA integrates Retrieval-Augmented Generation (RAG), a Graph-of-Thought (GoT) planning mechanism, and a reflection loop guided by the Lean compiler. It also introduces AriaScorer, a semantic checker that retrieves term definitions from Mathlib for grounding-based verification. Experiments show improvements over prior baselines.

**Strengths:**

1. The paper clearly motivates improving statement-level formalization before theorem proving by introducing ARIA, an agent designed to convert natural language mathematical statements into Lean 4 formalizations.

2. It presents a well-structured and competently engineered system that integrates Retrieval-Augmented Generation (RAG), a Graph-of-Thought (GoT) planning mechanism, and a reflection loop guided by the Lean compiler, alongside AriaScorer, a semantic checker leveraging Mathlib for grounding-based verification.

3.It evaluates the model on various benchmarks, including conjecture-level tasks.

**Weaknesses:**

1. The paper claims novelty by integrating RAG, GoT, and the reflection loop guided by the Lean compiler. However, each of these components already exists in prior work. ARIA merely combines these elements without introducing a new algorithmic principle or theoretical insight. This makes the paper engineering incremental rather than conceptually innovative
2. The core metric is defined using AriaScorer, a tool introduced in the same paper, which introduces a potential self-evaluation bias: the observed performance gains may reflect alignment with the metric rather than genuine capability. Furthermore, only the Conjectures dataset has been human-verified, leaving the main benchmarks unchecked. Alternative metrics for autoformalizers, such as typecheck and BEq [1], exist, and reporting results under these metrics would provide a more robust and credible evaluation.
3. Table 1 compares ARIA with Goedel-V2, although most results reported are for pass@1. However, “pass@k” sampling is not equivalent to multi-stage agentic reasoning. Additionally, regarding the Conjectures dataset, I would like to ask why ARIA exhibits such a significant advantage on this data. It seems likely that other models may not have encountered this type of data during training, resulting in 0% accuracy for those models.

[1] Qi Liu, Xinhao Zheng, Xudong Lu, Qinxiang Cao, Junchi Yan. Rethinking and Improving Autoformalization: Towards a Faithful Metric and a Dependency Retrieval-based Approach

**Questions:**

Please refer to the Weakness section.

---

> ### Author Response · Authors · 2025-11-24
> **Response to Reviewer ZfvQ (1/3)**
>
> **R W1: Clarification on Conceptual Insight and Architectural Novelty**
>
> We appreciate the reviewer's perspective on novelty. We respectfully claim that Aria offers a distinct conceptual contribution.
>
> First, our core insight is novel. Research-level mathematics often necessitates definitions that extend beyond the current scope of Mathlib. In dealing with such advanced mathematics, human mathematicians rely on hierarchical relationships to construct new concepts and theories. To the best of our knowledge, prior autoformalizers have not explicitly addressed the challenge of hierarchically constructing definitions.
>
> Second, our methodology is purposefully adapted to this insight, rather than being an arbitrary combination of techniques. Though the Graph-of-Thought (GoT) naturally aligns with the hierarchical intuition of mathematical definitions, to the best of our knowledge, no prior system has utilized this specific approach for formalization. While RAG and reflection techniques are indeed foundational to recent advancements, a single-layer RAG approach proves inadequate for conjecture-level problems. We observe that although basic RAG mitigates hallucinations for existing interfaces, it remains vulnerable when dealing with novel definitions: when the model cannot retrieve a relevant concept from Mathlib, it invariably resorts to hallucinating a non-existent interface. Our approach effectively resolves this issue. Our main results demonstrate a clear advantage over baselines. In Appendix A, we also present a comparison of examples generated by our model against other baselines. These examples clearly demonstrate the effectiveness of the GoT architecture in handling problems with complex definitions. Furthermore, as detailed in Appendix C.2, the critical role of GoT is particularly evident in the Conjecture dataset: without the GoT module, the system successfully formalized only a single conjecture, underscoring the necessity of our hierarchical approach.
>
> In summary, Aria offers conceptual insight and a novel hierarchical architecture, contributing to research-level auto-formalization.
>
> **R W2: Rationale for AriaScorer and Limitations of Syntactic Metrics**
>
> We sincerely thank the reviewer for ensuring evaluation rigor. We address the concerns regarding metric validity as follows:
>
> 1. **Addressing Self-Evaluation Bias:** Crucially, AriaScorer is not utilized during generation or reflection. The Aria agent relies solely on compiler feedback. AriaScorer is applied strictly post-hoc, ensuring that observed gains reflect genuine improvements, not optimization towards the metric.
>
> 2. **Validation against Human-Verified Results:** AriaScorer was explicitly designed to function as a reliable and independent automatic evaluator. To ensure its validity, we performed manual inspections of its outputs to verify strict alignment with human judgment. We constructed an expert-verified ground truth dataset for FATE-X (which is mentioned in Section 4.3.1). By validating AriaScorer against human annotations on Aria's outputs, we confirmed the reliability of our metric (shown in Table 2), thereby establishing the credibility of our evaluation results. We believe the rigorous manual verification conducted on both the Conjecture dataset and the FATE-X results sufficiently demonstrates the system's effectiveness and the Scorer's alignment with human judgment.
>
> 3. **Why Syntactic Metrics (BEq, typecheck, etc.) Was Not Used:** While methods like BEq and typecheck are standard for simpler tasks, we find it ill-suited for research-level mathematics. At this level, proving logical equivalence between a statement with synthesized definitions and a reference formal statement is often non-trivial. This limitation causes BEq to erroneously reject valid formalizations, resulting in false negatives. To illustrate why BEq fails in this domain, we present a specific case involving the E8 Kleinian Singularity:
>
> This is the code generated by Aria:
>
> ```lean
> import Mathlib
> noncomputable def kleinian_singularity_E8_polynomial : MvPolynomial (Fin 3) ℂ :=
>   (MvPolynomial.X 0) ^ 2 + (MvPolynomial.X 1) ^ 3 + (MvPolynomial.X 2) ^ 7
> abbrev E8_singularity_quotient_ring :=
>   (MvPolynomial (Fin 3) ℂ) ⧸ (Ideal.span {kleinian_singularity_E8_polynomial})
> instance kleinian_singularity_E8_ideal_isPrime :
>   (Ideal.span {kleinian_singularity_E8_polynomial}).IsPrime := by sorry
> theorem isUFD_E8_singularity_quotient_ring :
>   UniqueFactorizationMonoid E8_singularity_quotient_ring := by sorry
> ```
>
> And this is the reference code provided in FATE-X dataset:
>
> ```lean
> import Mathlib
> /--
> The ring $R = \\mathbb{C}[x, y, z] / (x^2 + y^3 + z^7)$.
> -/
> abbrev R : Type := (MvPolynomial (Fin 3) ℂ) ⧸ Ideal.span {(.X 0 ^ 2 + .X 1 ^ 3 + .X 2 ^ 7 : MvPolynomial (Fin 3) ℂ)}
> /--
> $\\mathbb{C}[x, y, z] / (x^2 + y^3 + z^7)$ is a UFD.
> -/
> theorem quotient_not_UFD :
>   ∃ (h : IsDomain R),
>     (UniqueFactorizationMonoid R) := by sorry
> ```

---

> > ### Author Response · Authors · 2025-11-24
> > **Response to Reviewer ZfvQ (2/3)**
> >
> > (cont.)
> >
> > First, we claim that both theorems assert the same fact: the coordinate ring of the E8 singularity is a Unique Factorization Domain (UFD).
> >
> > To formally prove that these two statements are equivalent in Lean (and thus satisfy a strict checker, e.g. BEq), one would need to perform three non-trivial steps:
> >
> > - Ring Identification: Prove `R = E8_singularity_quotient_ring`. This requires unfolding the definitions, returning to the quotient construction, and applying rewrite tactics.
> >
> > - Domain Verification: Prove that `R` is an integral domain. This follows from the `kleinian_singularity_E8_ideal_isPrime` instance, but requires a non-trivial proof step.
> >
> > - Logical Elimination: Prove the lemma `(∃ (h : IsDomain R), UniqueFactorizationMonoid R) ↔ UniqueFactorizationMonoid R`. This involves existential elimination logic.
> >
> > In summary, while the equivalence proof is not impossible, it requires unfolding 4-5 technical layers and employing intermediate-level Lean tactics involving quotient structure equalities and type class constructions. BEq operates on structural equality and cannot bridge this gap.
> >
> > This example exemplifies a broader challenge in research-level formalization: natural language equivalents frequently diverge into distinct Lean definitions due to specific implementation choices.The specific reasons contributing to the non-triviality of equivalence in Lean include:
> >
> > - Multiple Mathematical Definitions: A single concept often has multiple equivalent mathematical definitions. Different contexts (or authors) may prefer different formulations, leading to distinct structures or type classes in Lean.
> >
> > - Bundled Type Classes: Structures can be "bundled" differently. For example, a two-variable polynomial ring can be formalized as `MvPolynomial (Fin 2) R` or `Polynomial (Polynomial R)`. These are not definitionally equal; proving their equivalence requires constructing a complex algebraic isomorphism (`≃ₐ[R]`).
> >
> > - Inheritance Structures (Diamonds): Type classes inherit from others. Different inheritance paths can lead to "diamond" problems where the formal representations diverge despite representing the same object. Notably, this limitation is explicitly recognized in the foundational work on BEq by Liu et al. (2025). In Appendix A.4 (Failure Case Analysis, Figure 5) of their paper "Rethinking and Improving Autoformalization", they admit that BEq fails to resolve such inheritance divergences. This issue is acute in our framework: the Graph of Thoughts (GoT) planner synthesizes deeply structured, multi-layer definition chains. This structured approach naturally induces diamond patterns.
> >
> > These complexities render syntactic metrics like BEq inadequate, necessitating AriaScorer for plausible, semantic-level verification.
> >
> > In summary, AriaScorer ensures reliability through strict architectural decoupling to suppress potential hallucinations, and its precision is validated by human experts to reach 95.5% in our chosen setting. Given the demonstrated inability of syntactic metrics like BEq to capture semantic equivalence in research-level mathematics, we maintain that AriaScorer represents a necessary, reliable, and unbiased standard for this domain, providing a faithful measure of the system's true formalization capabilities.

---

> > > ### Author Response · Authors · 2025-11-24
> > > **Response to Reviewer ZfvQ (3/3)**
> > >
> > > **R W3: Comparison Rationale and Performance Attribution**
> > >
> > > We thank the reviewer for these questions regarding fair comparison and performance attribution.
> > >
> > > We agree that agentic reasoning differs from sampling. Our objective is to demonstrate that system design outperforms brute-force sampling (pass@k). Furthermore, Goedel-V2 is neither designed for nor supports iterative reflection and hierarchical decomposition. The comparison with Goedel-V2 at pass@128 highlights the superiority of our system design over brute-force scaling. Despite starting with comparable single-pass capabilities (Gemini pass@1 vs. Goedel-V2 pass@1), Aria attains higher accuracy with significantly fewer model calls, validating the efficiency of our structured approach. This validates our novel graph-based and reflection-based architecture as a powerful framework for complex formalization.
> > > Regarding Aria's significant advantage on the Conjectures dataset, we clarify that the performance gap stems from architectural capability rather than data contamination differential data exposure during pre-training. As shown in Appendix A.2, Gemini fails to translate the conjecture, hallucinating a non-existent interface "IsSystemOfParameters". Existing architectures struggle because research-level conjectures rely on nested definitions absent from Mathlib, requiring hierarchical decomposition to ground them. Without this mechanism, general LLMs (e.g., Gemini) inevitably hallucinate non-existent interfaces (e.g., IsRegularLocalRing, IsCohenMacaulayModule), while specialized models (e.g., Goedel) lack the semantic depth to comprehend such complexity, yielding incoherent outputs (as illustrated in Appendix A). Aria overcomes these barriers systematically: RAG mitigates hallucinations for existing library concepts, while GoT constructively synthesizes the missing high-level definitions. Furthermore, to manage the increased structural complexity introduced by GoT, we employ compiler reflection to guarantee validity. This combined approach is the key driver of our significant performance advantage on the Conjectures dataset.
> > >
> > > Finally, we respectfully clarify that the performance gap is not attributable to differential data exposure during pre-training. Notably, the base model (Gemini) without our pipeline performs similarly to Goedel-V2, also failing with 0% accuracy on conjectures. If the base model had simply "memorized" this data, the one-pass baseline would have achieved significantly higher scores; instead, it suffers from the same hallucination issues, confirming that the success is not due to inherent model knowledge. This is further corroborated by our ablation study in Appendix C, where removing any single component (e.g., GoT or Reflection) causes Aria's performance on the Conjecture dataset to drop to near zero. Crucially, we need to clarify that Aria is an inference pipeline utilizing a frozen base model, with no domain-specific fine-tuning conducted for these conjectures. Therefore, as detailed in the previous paragraph, the decisive factor is not specific data exposure, but rather our structural approach.

---

### Official Review · Reviewer_vbY2 · 2025-10-31

**Soundness:** 3
**Presentation:** 3
**Contribution:** 3
**Rating:** 4
**Confidence:** 5

**Summary:**

The paper proposes **ARIA**, an agent for translating informal mathematical statements into Lean code through **Graph-of-Thought (GoT)** reasoning, **retrieval-augmented generation (RAG)** from Mathlib, and a **compiler-in-the-loop** feedback mechanism. It also introduces **AriaScorer**, a term-grounded semantic checker ensuring meaning consistency between informal and formal statements. Experiments on **ProofNet**, **FATE**, and real conjectures show **state-of-the-art** performance, demonstrating ARIA’s ability to handle complex, research-level formalization tasks.

**Strengths:**

- Introduces a **dependency-graph (GoT)** framework that mirrors human reasoning, enabling structured decomposition and synthesis of unseen mathematical concepts.
- Combines **RAG grounding**, **GoT planning**, and **compiler reflection** into a robust, self-correcting pipeline, with solid ablation evidence.
- Well-structured and clearly illustrated; figures and examples effectively explain the dependency graph process.
- Achieves **research-level formalization** beyond prior methods and improves semantic reliability through **term-level grounding** (AriaScorer).

**Weaknesses:**

- The paper does not provide the detailed prompts used in the key stages—decomposition, grounding, synthesis, and reflection. Since the entire pipeline relies on prompt-driven reasoning, the absence of these examples makes it difficult to reproduce the workflow or evaluate design choices. Including representative prompts or templates would significantly improve clarity and reproducibility.

- All experiments are conducted on datasets from algebra and commutative algebra, limiting the demonstrated generality of ARIA. It remains unclear whether the dependency-graph and grounding strategies would perform equally well in other mathematical domains such as analysis, topology, or geometry. Broader testing or discussion on domain adaptation would strengthen the paper’s scope and impact.

- The LeanSearch component relies on a locally indexed Mathlib database, which introduces significant computational and storage overhead. A discussion of indexing efficiency, caching, or alternative lightweight retrieval strategies would make the system more practical for large-scale or real-time deployment.

**Questions:**

Please refer to Weaknesses.

---

> ### Author Response · Authors · 2025-11-24
> **Response to Reviewer vbY2 (1/2)**
>
> **R W1: Inclusion of Full Prompt Templates**
>
> We thank the reviewer for suggestion. We agree that detailed prompts are essential for reproducibility and evaluating our design choices. Due to space constraints, we were unable to include the full prompt set in the comments. We have now provided the complete templates in Appendix D of the revision.
>
> **R W2: Domain Generalization**
>
> We appreciate the reviewer’s valid concern regarding domain generalization. Indeed, we primarily test our pipeline on the algebra dataset FATE. However, we believe the results on Proofnet not only enable comparing our results with other methods, but also showcase our pipeline's capabilities in various domains. The ProofNet benchmark validated our agent on a broader range of undergraduate mathematics, including number theory, analysis and topology. This demonstrates that Aria's mechanisms are not strictly limited to Algebra.
>
> Specifically, we conducted a breakdown of the ProofNet benchmark by domain. As shown in the table below, Aria achieves highly consistent performance across these diverse fields. Furthermore, Aria surpasses the strongest baseline (Goedel-V2) in every subfield.
>
> | Metric | Algebra | Analysis | Number Theory | Topology |
> | :--- | :---: | :---: | :---: | :---: |
> | **Aria (Ours) Compiler** | **97.4%** | **100%** | **100%** | **96.7%** |
> | **Aria (Ours) Final Acc.** | **64.7%** | **64.8%** | **71.4%** | **56.7%** |
> | Goedel Compiler | 54.7% | 81.8% | 90.5% | 26.7% |
> | Goedel Final Acc. | 28.9% | 44.3% | 47.6% | 11.7% |
>
> Our concentration on algebra was driven by the absence of benchmark series in other domains that offer a progressive difficulty gradient comparable to the combination of FATE. To our knowledge, a comparable progressive benchmark suite does not yet exist for fields like advanced analysis. This specific difficulty progression is crucial, as it allows us to demonstrate that Aria's advantage over baselines widens as problem complexity and dependency depth increase. Furthermore, the specific subset of 14 conjectures constitutes a distinct high-difficulty tier, which is rare and challenging to benchmark across mathematical subfields.
>
> On the other hand, the decision to anchor our experiments in Algebra is rooted in the maturity of the theorem and definition infrastructure in Mathlib. This infrastructure for Algebra is exceptionally well-developed compared with other domains, offering the rich formalized grounding necessary to enable the formalization of modern mathematical concepts.
>
> Crucially, research-level mathematics in most domains is characterized by deep hierarchy of definitions and hierarchical conceptual dependencies. Commutative algebra not only exemplifies this intrinsic complexity but also possesses the sufficient infrastructure to support it. Consequently, this domain serves as an ideal testbed for demonstrating progress toward conjecture-level formalization, aligning with the core objectives of our study.
>
> We note that domains with less developed infrastructure pose significant barriers to conjecture formalization, as fundamental prerequisite concepts are often missing from the library. Despite the difficulty of locating frontier conjectures in domains with sufficient library support, we regard the reviewer's suggestion as highly valuable. Therefore, we dedicated effort to identifying a suitable candidate and successfully located a conjecture in topology, where our system demonstrated robust performance. Here we present the formalization of the conjecture (Borel's Conjecture).

---

> > ### Comment · Reviewer_vbY2 · 2025-11-26
> >
> > I now understand your motivation for choosing the algebra dataset. However, we don’t always have a well-developed domain infrastructure that allows us to automatically retrieve existing meta-tactics. In such cases, the generalizability of this approach seems somewhat limited.

---

> > > ### Author Response · Authors · 2025-11-28
> > > **Response to Reviewer vbY2**
> > >
> > > We thank the reviewer for this comment. We respectfully clarify that our system focuses on **autoformalization** (generating formal definitions and statements) rather than proof generation. Therefore, our retrieval module is designed to target Mathlib **definitions**, rather than the meta-tactics typically used in automated theorem proving.
> > >
> > > Nonetheless, we fully acknowledge the reviewer's underlying concern: the maturity of definition infrastructure indeed varies across domains, and these gaps pose a significant challenge for autoformalization. Crucially, however, we observe that even in less developed domains (e.g., topology, analysis), the fundamental building blocks remain universally available in Mathlib. This is because all high-level definitions, regardless of domain maturity, are ultimately composed of core logical and set-theoretic primitives. To leverage this universal availability, Aria advances beyond standard single-retrieval RAG (which rely on retrieving an exact match and fail when infrastructure is sparse). By employing a Graph of Thoughts (GoT) architecture, the system recursively decomposes complex concepts into these always-available components, thereby decoupling performance from the maturity of the domain library.
> > >
> > > Such autonomous construction is most decisive when addressing research-level conjectures, the primary focus of our work. This is highlighted in the Borel's Conjecture case study presented in our response to W2. In this specific instance, Mathlib lacked the high-level definitions for `IsClosedManifold` and `IsAsphericalTopologicalManifold`.  A standard RAG approach would fail by hallucinating these interfaces to bridge the gap. In contrast, Aria autonomously constructed these missing concepts by assembling available primitives (e.g., `IsManifold`, `CompactSpace`, `PathConnectedSpace`), and successfully formalized the conjecture.
> > >
> > > Our experimental results from the ProofNet benchmark can also support this claim. As illustrated in the domain breakdown table in our response to W2, Aria maintains consistent performance across diverse fields, including analysis, number theory, and topology. This consistency indicates that Aria is resilient to variations in library maturity, confirming the cross-domain robustness of the GoT architecture.
> > >
> > > Collectively, our experiments and analyses confirm that Aria is not limited by the absence of domain-specific infrastructure. By grounding high-level concepts into basic primitives, our approach remains robust even in domains with sparse definition libraries.

---

> ### Author Response · Authors · 2025-11-24
> **Response to Reviewer vbY2 (2/2)**
>
> (cont.)
>
> ```lean
> import Mathlib
>
> /-- A closed manifold is a compact manifold with empty boundary. -/
> class IsClosedManifold {𝕜 : Type*} [NontriviallyNormedField 𝕜] {E : Type*}
>   [NormedAddCommGroup E] [NormedSpace 𝕜 E] {H : Type*} [TopologicalSpace H]
>   (I : ModelWithCorners 𝕜 E H) (n : WithTop ℕ∞) (M : Type*) [TopologicalSpace M]
>   [ChartedSpace H M] extends IsManifold I n M, CompactSpace M : Prop where
>   /-- The boundary of a closed manifold is empty. -/
>   boundaryless : {x : M | I.IsBoundaryPoint (chartAt H x x)} = ∅
>
>
> /-- An aspherical topological manifold is a topological manifold `M` that is path-connected and for which the `k`-th homotopy group `πₖ(M, x)` is trivial for all `k ≥ 2` and all basepoints `x : M`. -/
> structure IsAsphericalTopologicalManifold
>     {𝕜 : Type*} [NontriviallyNormedField 𝕜]
>     {E : Type*} [NormedAddCommGroup E] [NormedSpace 𝕜 E]
>     {H : Type*} [TopologicalSpace H] (I : ModelWithCorners 𝕜 E H)
>     (n : WithTop ℕ∞) (M : Type*) [TopologicalSpace M] [ChartedSpace H M] : Prop where
>   /-- An aspherical topological manifold is a topological manifold. -/
>   is_manifold : IsManifold I n M
>   /-- An aspherical topological manifold is path-connected. -/
>   path_connected : PathConnectedSpace M
>   /-- The `k`-th homotopy group of an aspherical topological manifold is trivial for `k ≥ 2`. -/
>   homotopy_groups_trivial (k : ℕ) (hk : 2 ≤ k) (x : M) :
>     Subsingleton (HomotopyGroup (Fin k) M x)
>
>
> /-- Let $M$ and $N$ be closed and aspherical topological manifolds. If $f \colon M \to N$ is a homotopy equivalence, then $f$ is homotopic to a homeomorphism. -/
> theorem borel_conjecture_for_topological_manifolds
>     {𝕜 : Type*} [NontriviallyNormedField 𝕜]
>     {E : Type*} [NormedAddCommGroup E] [NormedSpace 𝕜 E]
>     {H : Type*} [TopologicalSpace H]
>     {I : ModelWithCorners 𝕜 E H}
>     {n : WithTop ℕ∞}
>     {M : Type*} [TopologicalSpace M] [ChartedSpace H M]
>     [IsClosedManifold I n M]
>     (hM_aspherical : IsAsphericalTopologicalManifold I n M)
>     {N : Type*} [TopologicalSpace N] [ChartedSpace H N]
>     [IsClosedManifold I n N]
>     (hN_aspherical : IsAsphericalTopologicalManifold I n N)
>     (f : ContinuousMap.HomotopyEquiv M N) :
>     ∃ (g : M ≃ₜ N), ContinuousMap.Homotopic f.toFun (g : C(M, N)) := by
>   sorry
> ```
>
> **R W3: Runtime Analysis of LeanSearch**
>
> We appreciate the reviewer’s focus on system efficiency. To address this, we analyzed the specific latency overhead introduced by LeanSearch.
>
> The time consumed by LeanSearch is negligible compared to the total inference time. The main computational overhead comes from indexing the entire library, which is a one-time offline process. In our setup, this requires only about 3 hours on a single GPU. Once indexed, the online phase involves merely embedding the single query item and performing the efficient HNSW calculation.
>
> During the formalization of a single problem, the retrieval process involves two steps: embedding the query and performing an HNSW (Hierarchical Navigable Small World) search. Embedding a single theorem statement takes approximately 30 ms, Given that the Mathlib corpus contains approximately 300,000 entries, HNSW via Chroma DB allow for nearest neighbor retrieval in mere milliseconds. Consequently, the total retrieval latency constitutes a minimal fraction (< 1%) of the overall runtime, which is dominated by the LLM's token generation.
>
> Moreover, LeanSearch is open-source and provides a ready-to-use API. This makes it easy to integrate into other systems, removing engineering barriers.

---

> > ### Comment · Reviewer_vbY2 · 2025-11-26
> >
> > It seems that only the **Prompt for the Decomposition Phase** provides a one-shot prompt. Does the whole process rely heavily on prompt design? It’s missing an ablation study on prompt sensitivity, as well as experiments on few-shot learning.

---

> > > ### Author Response · Authors · 2025-11-28
> > > **Response to Reviewer vbY2**
> > >
> > > We agree with the reviewer that prompt design plays a important role in the success of agentic pipelines. Our current prompt design stabilized after a series of iterative experiments during our early development phase. Below, we detail this evolution process to explain our rationale for the current design.
> > >
> > > In the early stages, we paid close attention to the impact of prompt design to ensure the system functioned as intended. We explicitly structured the tasks to standardize its behavior. For example, in the Reflection Module, we enforced a strict output sequence requiring an "Analysis and Correction Plan" prior to the "Corrected Lean 4 Code". For instance, in the statement formalization module, we enforced a strict output constraint requiring the model to generate only the theorem statement terminated by `sorry`, explicitly prohibiting the generation of proofs. This constraint prevents the model from generating potential errors in the proof body, ensuring that compilation feedback reflects only the correctness of the statement itself. However, we identified a unique challenge specific to the Decomposition Phase. Unlike other modules where explicit instructions suffice, this phase entails inherent structural ambiguity regarding the intended granularity (i.e., determining the exact canonical and atomic level at which a concept should be split). For instance, when decomposing "Non-zero principal prime ideal", the model would often generate insufficiently decomposed nodes like "principal prime ideal" or "non-zero prime ideal". This behavior is suboptimal because it fails to achieve sufficient complexity reduction: these intermediate nodes still combine multiple attributes rather than atomic attributes (i.e. "principal ideal", "non-zero ideal", "prime ideal"). To correct this and force the direct decomposition into atomic, canonical units, we introduced representative examples to concretize the abstract criteria.
> > >
> > > We also experimented with providing additional examples at each stage, but observed only **marginal improvements**. Moreover, Adding examples increased the context length, directly resulting in higher inference costs. For cost-effectiveness, we pruned the prompt to retain only the essential components. More critically, once structured execution capability was established, the analysis of remaining failures exposed deeper domain knowledge gaps. Specifically, the model frequently struggled to construct complex, deeply nested Lean terms or failed to accurately recall specific Mathlib interfaces. We realized that these issues exceeded the scope of what in-context examples could resolve. Accordingly, we redirected our optimization efforts from prompt engineering toward architectural improvements, such as enhancing the clarity of supportive information returned by the Retrieval and Verification modules. Additionally, adding examples increased the context length, directly resulting in higher inference costs. For cost-effectiveness, we pruned the prompt to retain only the essential components.
> > >
> > > More broadly, we posit that unlike earlier generations of smaller LLMs, modern powerful models (e.g. Gemini-2.5-pro) require only concise prompting and examples to establish sufficient task alignment. Hence, the primary focus of optimization in our work has shifted from Prompt Engineering to Context Engineering. Our architecture reflects this paradigm: the GoT framework decomposes complexity to minimize the cognitive load of individual tasks (creating focused, accurate contexts), while Retrieval and Reflection mechanisms augment the context with precise domain knowledge (increasing context quality).

---

### Official Review · Reviewer_YbKa · 2025-11-01

**Soundness:** 3
**Presentation:** 3
**Contribution:** 4
**Rating:** 8
**Confidence:** 4

**Summary:**

The authors introduce two contributions, Aria, an autoformalization method/pipeline which decomposes and grounds an informal statement in Mathlib to ensure correctness, and AriaScore, a new metric which improves upon other alignment scorers such as LeanScore and back-translation. The authors conduct experiments to demonstrate the effectiveness of their contributions.

**Strengths:**

Both AriaScore and Aria are significant contributions to the community. The authors conduct experiments and error analyses to show that AriaScore performs better at measuring alignment than previous metrics. The authors also show that each of their three components in Aria are important to high performance as shown by an ablation study. Overall, the authors do a good job supporting their claims and have a strong contribution.

**Weaknesses:**

I think it's worth distinguishing this approach from papers such as (Liu et al, 2025), which also do RAG autoformalization.

An error analysis of the components of Aria would be very useful to understanding how this method improves. The authors make a claim (~L361) that hallucination of incorrect interfaces is the main source of error. However, why does a single-retrieval RAG system not accomplish a similar improvement in that case? Or perhaps it does, but just not to the same degree? Your ablation study answers this to some extent. But I believe an error analysis would be most beneficial.

**Questions:**

How does your system know the **informal** definitions of things it can't find in Mathlib, e.g., Cohen-Macaulay Module? My understanding is that this would be necessary information to be able to continue breaking down the definition into grounded terms, but I'm not clear on how this works.

While it's not in autoformalization, a similar technique as your "reflection" module has been used in similar fields such as automated theorem proving (see: COPRA from Thakur et al. 2024) among others. Maybe worth mentioning in related works.

---

> ### Author Response · Authors · 2025-11-24
> **Response to Reviewer YbKa**
>
> **R W1: Distinction from previous RAG method**
>
> We thank the reviewer for this insightful comparison. We explain the specific improvements of our work over prior RAG approaches as follows.
>
> While both Aria and recent works like Liu et al. (2025) utilize RAG, our primary contribution lies in the application of the Graph of Thoughts (GoT) algorithm to transcend the limitations of standard retrieval approaches.
>
> Prior RAG-based approaches typically function as linear retrieval-augmented generation processes. They lack a mechanism for structured, multi-turn reflection and are often unable to construct deeply nested definitions. Consequently, for absent concepts requiring deep nesting, single-pass RAG compels the model to output a complex chain of definitions in a single inference step. This approach fails to leverage the model's full reasoning capabilities for structural decomposition, resulting in flawed formalizations.
>
> **R W2: Error Attribution: Single-Retrieval RAG vs. Aria**
>
> This is an excellent suggestion. We agree that detailed error analysis is vital for isolating the source of improvements. Mechanistically, a single-retrieval RAG performs adequately when all necessary interfaces already exist in Mathlib. However, it hits a hard limit when a concept is absent from the library (e.g., `Cohen-Macaulay Ring`). Upon failing to retrieve a matching definition, the model invariably tends to hallucinate a non-existent interface (e.g., `IsCohenMacaulayRing`) to complete the code, as it lacks the mechanism to recognize this retrieval void.
>
> To quantify this impact, we conducted an error attribution analysis on the Conjectures dataset, focusing specifically on failures caused by hallucinated interfaces. The without GoT version failed on 13 out of 14 problems, and all of these failures were attributed to the hallucination of missing interfaces. By introducing the GoT pipeline, only 6 (out of 8 total failed cases) failure cases were attributed to interface issues. Specifically, these represent cases where the synthesis of the requisite intermediate definition was attempted but failed. This contrast demonstrates that while single-retrieval RAG is prone to hallucination by default when knowledge is missing, Aria effectively converts many of these potential hallucinations into valid, synthesized definitions.
>
> **R Q1: Source of Knowledge for Missing Concepts**
>
> We thank the reviewer for this insightful question.
>
> For recognizing and decomposing informal mathematical definitions, Aria relies on the prior knowledge already embedded in modern LLMs. As shown in Figure 1(b) of the FATE benchmark paper (Jiang et al., 2025), proprietary models such as DeepSeek-R1 perform poorly on formal proof generation yet achieve strong performance on the corresponding informal reasoning tasks, suggesting that LLMs generally have no difficulty understanding informal mathematical notions. In our own example (Figure 1), even without providing any description of `Cohen–Macaulay module`, Gemini-2.5-Pro correctly recalls and expands its standard definition. The same phenomenon appears for `system of parameters` in Example 2 of Appendix A.2.
>
> In addition, informal mathematical writing typically provides brief explanations for concepts that are not widely known, further breaking them down into forms that LLMs can reliably interpret. In practice, we rarely observe the LLM misunderstanding such informal notions. The primary difficulty in autoformalization therefore does not lie in understanding the informal definition itself, but in constructing the precise Lean expressions that match it.
>
> **R Q2: Discussion on Related Work**
>
> We sincerely thank the reviewer for bringing this work to our attention. We agree that certain lines of research in auto theorem proving, such as COPRA (Thakur et al., 2024), share the core philosophy of using execution feedback for iterative refinement. We have added a paragraph in the Related Work section of the revised paper to cite and discuss these approaches.

---

### Author Response · Authors · 2025-12-04
**General Response (1/3)**

We sincerely thank all the reviewers for their insightful and high-quality feedback. Generally, this paper introduces an effective pipeline, enhancing the formalization performance on research and conjecture level mathematics in both the generation and verification stages.

For the two-stage Aria procedure, we have architected a powerful system to effectively formalize nested novel definitions, extending the capability of formalization systems far beyond the currently available math repository. This is made possible by the coordination of multiple expert-verified components - decomposer, retriever, keyword grounder, and terminal formalizer - all of which contribute to satisfactory results on real-world datasets, and are further validated by careful manual inspections and ablation studies.

For AriaScorer, we build upon the effective foundation of LeanScorer. However, we identified a limitation when addressing research-level statements: reliance on superficial textual similarity often fails to capture the precise semantics of specific Lean terms. To make it strict, robust and adaptive, we incorporate both informal and formal term-level information for fine-grained justification. **This scorer module is kept strictly isolated from any part of the Aria autoformalization pipeline to avoid potential bias.** Recognizing that ground truth is paramount, we conducted a comprehensive and meticulous expert review of the generations in critical subsets, empirically validating the scorer's reliability and fidelity to human judgment.

We are gratified that the reviewers recognized Aria as a well-engineered system addressing the frontier of research-level autoformalization and acknowledged AriaScorer as a meaningful contribution to semantic verification. We are deeply grateful for the reviewers' encouraging remarks regarding Aria's motivation and logical coherence. We sincerely appreciate the recognition that our Graph-of-Thought (GoT) framework mirrors human reasoning to tackle definition synthesis, and that our robust, self-correcting pipeline achieving breakthrough results on conjectures constitutes a meaningful contribution to the community.

The constructive discussion during the rebuttal phase has been instrumental in refining our manuscript. In this response, we synthesize how the reviewers' insightful inquiries have guided us to clarify our methodological contributions and strengthen our empirical evidence. Reflecting these constructive exchanges, we have updated the manuscript to include discussions on the verification strategy of AriaScorer, deeper analyses of the GoT planner's ablation results, and a broader evaluation of domain generalization. Additionally, we have provided comprehensive prompt templates and detailed justifications regarding the limitations of syntactic metrics.

Below, we summarize some general issues raised by the reviewers.

1. **Why does our Aria formalization pipeline work and what makes it distinct from all previous systems?**

The reviewers advised clarifying the system's conceptual distinctiveness from three different aspects. Firstly, regarding architectural design, Reviewer ZfvQ questioned whether combining RAG, GoT, and Reflection constitutes a conceptual innovation or engineering integration. Reviewer YbKa recommended distinguishing our approach from standard RAG methods (e.g., Liu et al., 2025). Secondly, concerning performance and capability, Reviewer tTzy requested empirical evidence of synthesized novel definitions. And Reviewer vbY2 inquired about the generalizability of our approach across diverse mathematical domains. Most constructively, Reviewer Tkra probed the deep mechanism behind the "high compilation, low accuracy" phenomenon in our ablation studies, prompting us to elaborate on the core value of the GoT planner. These insightful questions and suggestions inspired us to articulate the effectiveness of our pipeline.

First, Aria is specifically targeted towards research-level statements. Previous methods fail to effectively formalize such problems due to their inability to manage hierarchically nested concepts. Human mathematicians rely on hierarchical relationships to construct new concepts when tackling such mathematics, and we designed GoT to align with this hierarchical intuition. Unlike single-layer RAG methods, where the absence of a dependent concept in Mathlib prevents the retrieval system from yielding a valid definition, thereby causing the LLM to invariably hallucinate a non-existent interface, Aria effectively resolves this issue through recursive synthesis. Comparative performance is detailed in Appendix A, and Appendix C.2 illustrates the critical role of GoT on the Conjectures dataset.

---

> ### Author Response · Authors · 2025-12-04
> **General Response (2/3)**
>
> (cont.)
>
> Regarding performance and capability, we addressed Reviewer tTzy's request by manually inspecting the synthesized definitions, confirming that specific definitions were indeed absent from the Mathlib commit. We have included two representative examples in our detailed response to substantiate their novelty. The performance gap on the Conjectures dataset (Aria 42.9% vs. Baselines 0%) provides strong empirical backing that this novel synthesis capability is both effective and essential for research-level formalization. Furthermore, Aria's performance advantage widens progressively across benchmarks of increasing complexity, from FATE-H to FATE-X, and finally to the Conjectures dataset, which confirms its effectiveness in handling deeply nested dependencies. Addressing Reviewer vbY2's concern regarding our concentration on algebra, we substantiated Aria's cross-domain robustness. We provided a domain breakdown of the ProofNet benchmark, demonstrating that Aria achieves consistent, superior performance across analysis, number theory, and topology, surpassing baselines in every subfield. We also formalized a topological conjecture (Borel's Conjecture), validating the effectiveness of GoT across different domains. These results confirm that Aria's efficacy is not limited to algebra but extends to any domain where fundamental logical primitives are available.
>
> Most constructively, Reviewer Tkra probed the deep mechanism behind the "high compilation, low accuracy" anomaly in our ablation studies. This inquiry prompted us to elaborate on the core value of the GoT planner. We clarified that this phenomenon reveals a fundamental trade-off between syntactic simplicity and semantic precision. Without GoT, the agent defaults to mapping complex concepts to approximate existing library terms. While these terms compile easily, they frequently fail to capture rigorous constraints. Conversely, GoT prioritizes semantic explicitness by enforcing the generation of rigorous, custom definitions. As problem complexity increases, this compilation gap diminishes while GoT's accuracy advantage widens significantly. This analysis confirms that the core value of GoT lies in unlocking the LLM's capacity for complex structural reasoning, a capability essential for research-level formalization where static retrieval fails.
>
> 2. **How to ensure our semantic verifier, AriaScorer, is appropriate, effective, and unbiased?**
>
> AriaScorer addresses the critical challenge of evaluating semantic consistency in autoformalization. Firstly, regarding appropriateness, Reviewer ZfvQ suggested alternative syntactic metrics (e.g., BEq [1]) for verification, while Reviewer Tkra inquired about the robustness and capability of AriaScorer's grounding mechanism. Secondly, regarding effectiveness, Reviewer Tkra questioned the iterative application of AriaScorer and the pipeline's capacity to avert semantic inconsistencies during formalization. Lastly, concerning unbiasedness, Reviewer ZfvQ raised valid concerns regarding potential self-evaluation bias, while Reviewer dKZk requested human evaluation to validate the main results.
>
> We appreciate the reviewers’ scrutiny regarding the verification methodology. Firstly, regarding the appropriateness of AriaScorer, we clarified that syntactic metrics (e.g., BEq [1]) are less applicable for research-level mathematics, because proving logical equivalence between a synthesized definition and a reference presents significant challenges at this level of complexity, leading BEq to erroneously reject valid formalizations. Following Reviewer ZfvQ's suggestion, we have added an analysis of BEq's applicability. Addressing Reviewer Tkra's concern regarding dataset completeness, we respectfully clarified that our grounding corpus is comprehensive. Furthermore, AriaScorer prioritizes authoritative formal definitions to override potential informal ambiguities, ensuring it remains an appropriate and robust checker for semantic consistency.
>
> Regarding iterative application and effectiveness, we addressed Reviewer Tkra's inquiry concerning the potential for semantic error propagation. We clarified that AriaScorer is applied strictly post-hoc to preclude self-evaluation bias. Motivated by the reviewer's valuable insight, we conducted a manual root cause analysis of all failure cases on FATE-X. This investigation confirmed that only a single instance of failure was attributable to a flaw in definition synthesis propagating to cause final semantic inconsistency. This empirical evidence validates the system's intrinsic robustness against error propagation, confirming that incorporating AriaScorer into the iterative loop is unnecessary.

---

> > ### Author Response · Authors · 2025-12-04
> > **General Response (3/3)**
> >
> > (cont.)
> >
> > Lastly, concerning unbiasedness, we addressed Reviewer ZfvQ's concern regarding self-evaluation bias and Reviewer dKZk's request for human evaluation. We clarified that AriaScorer is applied strictly post-hoc, decoupling evaluation from generation. To rigorously validate this, we constructed an expert-verified ground truth dataset for FATE-X. Achieving a precision of 95.5% against human experts, AriaScorer serves as a reliable, unbiased estimator of semantic consistency, confirming the validity of our reported main results.
> >
> >
> > **Conclusion**
> > Ultimately, through the integration of Aria and AriaScorer, we have empirically validated the system's robust application in Lean 4 environments, achieving compelling results on Conjecture benchmarks where prior systems failed. To advance this work further, the reviewers' constructive suggestions inspired us to enhance the system's usability and generalizability. As requested by Reviewer vbY2, we have included the full set of system prompts in Appendix D to facilitate immediate reproduction. Furthermore, inspired by the reviews, we have reflected on how this modular and effective framework can be adapted to even more diverse settings. For instance, addressing Reviewer vbY2's comment, we included an investigation of Aria across different mathematical domains in Appendix E, concretely establishing its cross-domain capabilities. Similarly, following the advice of Reviewer tTzy, we provided an architectural analysis on how Aria generalizes to other formal languages by simply swapping the compiler interface and retrieval corpus.
> >
> > Collectively, these comments have strengthened our confidence in and vision for the Aria system. In the future, we are also thrilled to see how this modular pipeline can be integrated into industrial data generation workflows, empowering models with spontaneous definition creation capabilities, and eventually serving as a catalyst for article and textbook-level auto-formalization across various domains and formal languages, benefiting the broader research community in mathematics and formalization.
> >
> >
> > [1] Qi Liu, Xinhao Zheng, Xudong Lu, Qinxiang Cao, Junchi Yan. Rethinking and Improving Autoformalization: Towards a Faithful Metric and a Dependency Retrieval-based Approach

---

### Meta-Review · Area_Chair_MQ4W · 2026-01-05

**Summary:**

The paper proposes a new auto-formalization pipeline based on statement decomposition and Mathlib definition retrieval. All reviewers recognize the effectiveness of the Aria pipeline, its individual components, and the AriaScore retrieval module.

The main concerns relate to generalization and novelty. Reviewers 1 and 2 question whether the approach generalizes to domains where concepts are missing in Mathlib for grounding. Reviewers 1 and 3 note similarities to existing RAG-style frameworks, with Reviewer 3 arguing that the combination of existing techniques limits the paper’s novelty.

Additional concerns include potential self-evaluation bias in AriaScore (Reviewer 3) and requests from Reviewers 3 and 5 for further results, such as type-checking accuracy, BEq, and human evaluation.

The authors’ rebuttal addresses the major concerns. It provides a detailed comparison with prior work, an example showing how Aria constructs missing concepts by assembling available primitives, additional manual verification, and clarification of why certain prior metrics are not suitable for this task. Based on the discussion, it is likely that two reviewers will raise their scores from borderline reject to positive.

Given the high likelihood of positive ratings from all five reviewers, I recommend acceptance.

**Reviewer Concerns:**

I believe the rebuttal addresses most of the reviewers’ concerns.

**Reviewer Scores:**

Reviewer 2 is very likely to raise their score, as they actively participated in the discussion and most concerns have been addressed, with the remaining points clarified in the authors’ follow-up responses.

Reviewer 3 may also raise the score, as the rebuttal provides comprehensive clarifications for each concern.

Reviewer 4 has already raised the score during the discussion.

---

### Decision · Program_Chairs · 2026-01-26

Accept (Poster)